



# Characterization of Brown Carbon absorption in different European environments through source contribution analysis

Hector Navarro-Barboza[1], Jordi Rovira[2], Vincenzo Obiso[1], Andrea Pozzer[3,4], Marta Via[2], Andres Alastuey[2], Xavier Querol[2], Noemi Perez[2], Marjan Savadkoohi[2,21], Gang Chen[5], Jesus Yus-Díez[6], Matic Ivancic[7], Martin Rigler[7], Konstantinos Eleftheriadis[8], Stergios Vratolis[8], Olga Zografou[8], Maria Gini[8], Benjamin Chazeau[9,20], Nicolas Marchand[9], Andre S.H. Prevot[9], Kaspar Dallenbach[9], Mikael Ehn[10], Krista Luoma[10,11], Tuukka Petäjä[10], Anna Tobler[12], Jaroslaw Necki[13], Minna Aurela[14], Hilkka Timonen[14], Jarkko Niemi[13], Olivier Favez[15], Jean-Eudes Petit[16], Jean-Philippe Putaud[17], Christoph Hueglin[18], Nicolas Pascal[19], Aurélien Chauvigné[19], Sébastien Conil[22], Marco Pandolfi[2], and Oriol Jorba[1]

[1]Barcelona Supercomputing Center, Plaça Eusebi Güell 1-3, Barcelona, 08034, Spain.

[2]Institute of Environmental Assessment and Water Research, c/Jordi-Girona 18-26, Barcelona, 08034, Spain.

[3]Atmospheric Chemistry Department, Max Planck Institute for Chemistry, Mainz, Germany.

[4]Climate and Atmosphere Research Center, The Cyprus Institute, Nicosia, Cyprus.

[5]MRC Centre for Environ. and Health, Environ. Research Group, Imperial College London, London, U.K.

[6]Centre for Atmospheric Research, University of Nova Gorica, Vipavska 11c, SI-5270 Ajdovščina, Slovenia.

[7]Aerosol d.o.o., Kamniška 39A, 1000 Ljubljana, Slovenia.

[8]ENRACT, Institute of Nuclear & Radiological Sciences & Technology, Energy & Safety, National Centre of Scientific Research "Demokritos", GR-15310 Ag. Paraskevi, Attica, Greece.

[9]Paul Scherrer Institute, PSI Center for Energy and Environmental Sciences.

[10]Institute for Atmospheric and Earth System Research/Physics, Faculty of Science, University of Helsinki, Helsinki, 00014, Finland.

[11]Atmospheric Research Centre of Eastern Finland, Finnish Meteorological Institute, Kuopio, Finland.

[12]Datalystica Ltd., Parkstrasse 1, 5234 Villigen, Switzerland.

[13]Helsinki Region Environmental Services Authority (HSY), 00240, Helsinki, Finland.

[14]Atmospheric Composition Research, Finnish Meteorological Institute, P.O. Box 503, 00101 Helsinki, Finland.

[15]Institut National de l'Environnement Industriel et des Risques (INERIS), 60550 Verneuil-en-Halatte, France.

[16]Laboratoire des Sciences du Climat et de l'Environnement, CEA CNRS UVSQ, Université Paris-Saclay, Gif-sur-Yvette, France.

[17]European Commission, Joint Research Centre (JRC), Ispra, Italy.

[18]Empa, Swiss Federal Laboratories for Materials Science and Technology, 8600 Duebendorf, Switzerland.

[19]Univ Lille, ICARE Data & Serv Ctr, CNRS, CNES,UMS 2877, Lille, France.

[20]Aix Marseille Univ, CNRS, LCE, Marseille, France.

[21]Department of Mining, Industrial and ICT Engineering (EMIT), Manresa School of Engineering (EPSEM), Universitat Politècnica de Catalunya (UPC), 08242 Manresa, Spain.

[22]Andra CMHM, DISTEC/EES, RD960, 55290 Bure, France.

**Correspondence:** Hector Navarro-Barboza (hector.navarro@bsc.es), Oriol Jorba (oriol.jorba@bsc.es)

**Abstract.**

Brown carbon (BrC) is a fraction of Organic Aerosols (OA) that absorbs radiation in the ultraviolet and short visible wavelengths. Its contribution to radiative forcing is uncertain due to limited knowledge of its imaginary refractive index ($k$). This





study investigates the variability of $k$ for OA from wildfires, residential, shipping, and traffic emission sources over Europe.
The MONARCH atmospheric chemistry model simulated OA concentrations and source contributions, feeding an offline optical tool to constrain $k$ values at 370 nm. The model was evaluated against OA mass concentrations from Aerosol Chemical Speciation Monitors (ACSM) and filter sample measurements, and aerosol light absorption measurements at 370 nm derived from Aethalometer[TM] from 12 sites across Europe. Results show that MONARCH captures the OA temporal variability across environments (regional, suburban and urban background). Residential emissions are a major OA source in colder months, while secondary organic aerosols (SOA) dominate in warmer periods. Traffic is a minor primary OA contributor. Biomass and coal combustion significantly influence OA absorption, with shipping emissions also notable near harbors. Optimizing $k$ values at 370 nm revealed significant variability in OA light absorption, influenced by emission sources and environmental conditions. Derived $k$ values for biomass burning (0.03 to 0.13), residential (0.008 to 0.13), shipping (0.005 to 0.08), and traffic (0.005 to 0.07) sources improved model representation of OA absorption compared to a constant $k$. Introducing such emission source-specific constraints is an innovative approach to enhance OA absorption in atmospheric models.

## 1 Introduction

Brown carbon (BrC) is the fraction of organic aerosols (OA) which exhibits light-absorbing properties, particularly in the ultraviolet and visible spectrum (Andreae and Gelencsér, 2006; Laskin et al., 2015). The role of BrC in atmospheric radiative forcing, while possibly significant, remains incompletely quantified (Brown et al., 2018; Zhang et al., 2020a; Sand et al., 2021). This is in part due to the historical treatment of OA in atmospheric models as mostly scattering solar radiation (Feng et al., 2013). BrC emissions originate from a variety of sources, significantly influenced by regional factors, including biomass, biofuel, and fossil fuel (Lu et al., 2015) combustion. Global estimates of BrC emissions reveal distinct regional patterns (Xiong et al., 2022). In Africa and South America, more than 70% of the primary BrC emissions are attributed to natural sources, such as wildfires. However, East Asia's BrC emissions are primarily anthropogenic, with residential solid fuel combustion accounting for more than 80% of the emissions. Europe presents a more mixed source profile, where natural sources are currently considered to be responsible for approximately 36% of BrC emissions, while residential activities (e.g., coal and solid biomass combustion for domestic heating) contribute approximately 48%. Furthermore, BrC is originating not only through primary emissions but also through secondary formation processes in the atmosphere, such as darkening (Kumar et al., 2018; Li et al., 2023). However, as BrC ages in the atmosphere, it can undergo photochemical and oxidative processes that lead to a reduction in its absorption capabilities, a phenomenon known as photobleaching (Hems et al., 2021). The complexity of BrC lies in its changing absorption characteristics and diverse composition. In fact, factors like burning conditions, solar exposure, and chemical composition determine BrC optical properties such as imaginary refractive index ($k$) and Absorption Angstrom Exponent (AAE). The optical properties of BrC present a high variability that arises from the formation of different chromophores and molecular structures, thus complicating the representation of BrC in climate models and our understanding of its atmospheric impact (Laskin et al., 2015; Brege et al., 2021; Washenfelder et al., 2022).



Several processes that alter BrC absorptive characteristics have recently been identified. Photobleaching stands out as a key process in this context. It describes the phenomenon in which BrC loses its absorptive capacity, particularly under conditions of high OH oxidation. For instance, a laboratory study conducted by Wong et al. (2019) observed that high-molecular weight BrC undergoes initial photoenhancement and subsequent gradual photobleaching, a phenomenon supported by observations from ambient BrC samples collected during fire seasons in Heraklion, Crete, Greece. Both primary and secondary BrC are subject to photobleaching, as shown in various studies, e.g. Forrister et al. (2015); Wang et al. (2018). Contrasting photobleaching, Kodros et al. (2020) introduced the concept of "dark aging" or darkening, a novel BrC secondary organic aerosol (SOA) formation pathway from biomass burning emissions. This process, which occurs under low or no light conditions and takes hours, involves OA oxidation initiated by the $NO_3$ radical with phenol, cresol, and furanoic compounds as primary reactants, and is influenced by relative humidity. Another process that has so far been less understood and is currently not treated in modeling studies is the *lensing effect* of BrC, as explored e.g. by Basnet et al. (2023); Cappa et al. (2012). This process can enhance the absorption of BrC. It is based on the mixing state of BrC, typically assuming a core-shell structure. In this structure, the core is composed of black carbon (BC), while the shell, which can consist of BrC or a purely scattering material, acts like a lens. This lens effect focuses and intensifies the light absorption of the entire particle, thereby potentially altering its radiative impact.

Understanding the optical properties of BrC is of fundamental importance, especially its *k* because is a key parameter used to quantify the light-absorbing properties of BrC. However, this parameter is not well constrained, and it exhibits significant uncertainty and variability. Research indicates that the imaginary refractive index of BrC can vary by 30-50%, highlighting the complex and diverse nature of its light-absorbing properties (Wang et al., 2013). Further studies, such as by Cheng et al. (2021), suggest that light-absorption properties of BrC vary depending on the source of emission, as demonstrated in comparisons of BrC generated from different fuels. Saleh (2020) introduced a classification for BrC based on its imaginary refractive index at $k_{550}$ into four categories ranging from very weakly ($10^{-4}$ to $10^{-3}$) to strongly absorbing ($> 10^{-1}$) BrC, with a noted correlation between BrC's source and its absorption category. This categorization highlights a gradient in absorption capabilities and associates more absorptive BrC, primarily resulting from high-temperature biomass combustion, with flatter absorption spectra. The classification underscores the complex and variable optical properties of BrC. It should be noted that the absorption categories for OA proposed by Saleh (2020) were based on 20 chamber experiments, which might not fully reflect the conditions found in the field.

A seminal work on modeling the effects of BrC in the atmosphere using 3D atmospheric models was the characterization of OA as absorbing with a $k_{550}$ assigned of 0.27 by Park et al. (2010). Feng et al. (2013) extended this by modeling the absorbing fraction of OA separated into two categories: moderately absorbing, characterized by a $k_{550}$ value of 0.003, and strongly absorbing, with a $k_{550}$ value of 0.03. Lin et al. (2014) continued this approach and classified into two groups: low ($k_{550}$: 0.001) and high ($k_{550}$:0.03). This classification, particularly distinguishing primary organic aerosol (POA) from SOA, was based on the understanding that POA generally exhibits greater absorption potential than SOA which is chemically very different. Saleh et al. (2014) advanced the field by deriving equations based on experimental data to parameterize the absorption of OA from



the ratio of emitted BC and OA. Such parameterization was subsequently utilized in the GEOS-Chem model by Saleh et al. (2015) and Wang et al. (2018), and in the CAM5 model by Brown et al. (2018) to represent OA absorption.

At a global scale, studies have highlighted the significant warming effect of BrC alongside BC and greenhouse gases (GHGs). Feng et al. (2013) estimated that BrC's warming effect could be approximately one-fourth that of BC, identifying BrC, mainly emitted from fuel combustion and open vegetation burning, as a substantial component contributing to global warming. Zhang
et al. (2020a) suggested that BrC might be a larger heating source in the tropical free troposphere than BC. A recent study by Liu et al. (2023) found that a decrease in single scattering albedo (SSA) at near-ultraviolet wavelengths significantly reduces the efficiency of the direct radiative forcing (DRF) due to strong absorption capabilities of BrC, which impacts both local and global radiation budgets. This is further supported by Wang et al. (2014), who utilized the GEOS-Chem model to demonstrate that incorporating BrC significantly improves the accuracy of absorption aerosol optical depth (AAOD) predictions by over
50% at AERONET stations, contributing an estimated +0.11 $Wm^{-2}$ to DRF.

Regional studies complement global models by offering detailed insights into BrC formation processes and its effects at a local scale, highlighting the importance of focusing on specific geographical areas to understand local and regional atmospheric phenomena. For example, research conducted in Northwestern and Southeastern Europe and northern peninsular Southeast Asia have revealed significant seasonal variations in BrC concentrations, which are particularly impactful in urban settings
often due to residential wood burning (Zhang et al., 2020b; Paraskevopoulou et al., 2023; Methymaki et al., 2023; Pani et al., 2021). These variations not only affect air quality but also complicate the broader understanding of regional climate impacts. Moreover, research conducted in the northwestern United States have examined the emissions of BrC from wildfires. These studies indicate that the inclusion of SOA formation and photobleaching effects in atmospheric models can enhance the simulation of aerosol optical properties. For example, incorporating these processes improves the representation of aerosol optical
depth (AOD) and single scattering albedo (SSA), making the model outputs more consistent with observed data (Neyestani and Saleh, 2022). These regional studies underscore the value of localized research in enhancing our understanding of BrC aerosols. They not only provide insights into the specific sources and behaviors of BrC in different environments but also help in refining global models that are used to predict atmospheric conditions. Unlike these studies, which often focus on specific events or short time periods, the current research extends over an entire year, providing a comprehensive analysis of seasonal
trends and identifying diverse BrC sources. This approach not only broadens the scope of understanding BrC dynamics but also enhances the predictive capabilities of climate models concerning BrC's environmental impacts.

In this research, we aim to investigate the light absorption properties of OA at different environments in Europe. We employ both modeling techniques and experimental approaches to constrain specific $k$ indexes for OA originating from different emission sources such as fires, residential, shipping, traffic, and others. We use the Multiscale Online Nonhydrostatic Atmosphere
Chemistry model (MONARCH; Badia and Jorba, 2015; Badia et al., 2017; Klose et al., 2021; Navarro-Barboza et al., 2024) to simulate the light absorption of OA in Europe during 2018. Our results provide an estimate of OA light absorbing properties in Europe. This comprehensive approach allows us to provide a first attempt to constrain OA optical properties representative of field conditions based on the current knowledge on emission sources and transport modeling. While previous extensive studies




in laboratory conditions such as Saleh (2020) are highly valuable for modeling purposes, they may not fully represent actual
ambient conditions.

In Section 2, we detail the experimental framework and the methodologies employed in this research. The observational
dataset is introduced and the modeling tools and optimization method to derive $k$ values described. Section 3 presents our
results, starting with an evaluation of the modeled OA mass against observational data and providing an in-depth analysis
of source contributions. We discuss then the optimization of $k$ values under different assumptions for OA emission sources.
Finally, Section 4 summarizes our key conclusions and outlines the recommendations arising from this study.

## 2  Materials and Methods

### 2.1  Observational dataset

The OA/Organic Carbon (OC) mass concentrations and multi-wavelengths absorption measurements used in this study were
collected at 12 sites in Europe covering urban, suburban, and regional background environments. Figure 1 and Table 1 show
the geographical locations of these atmospheric research stations. Figure 1 also shows the relative contribution of black carbon
(BC) and BrC to the total absorption measured at 370 nm that was obtained applying the procedure detailed in Section 2.1.1.





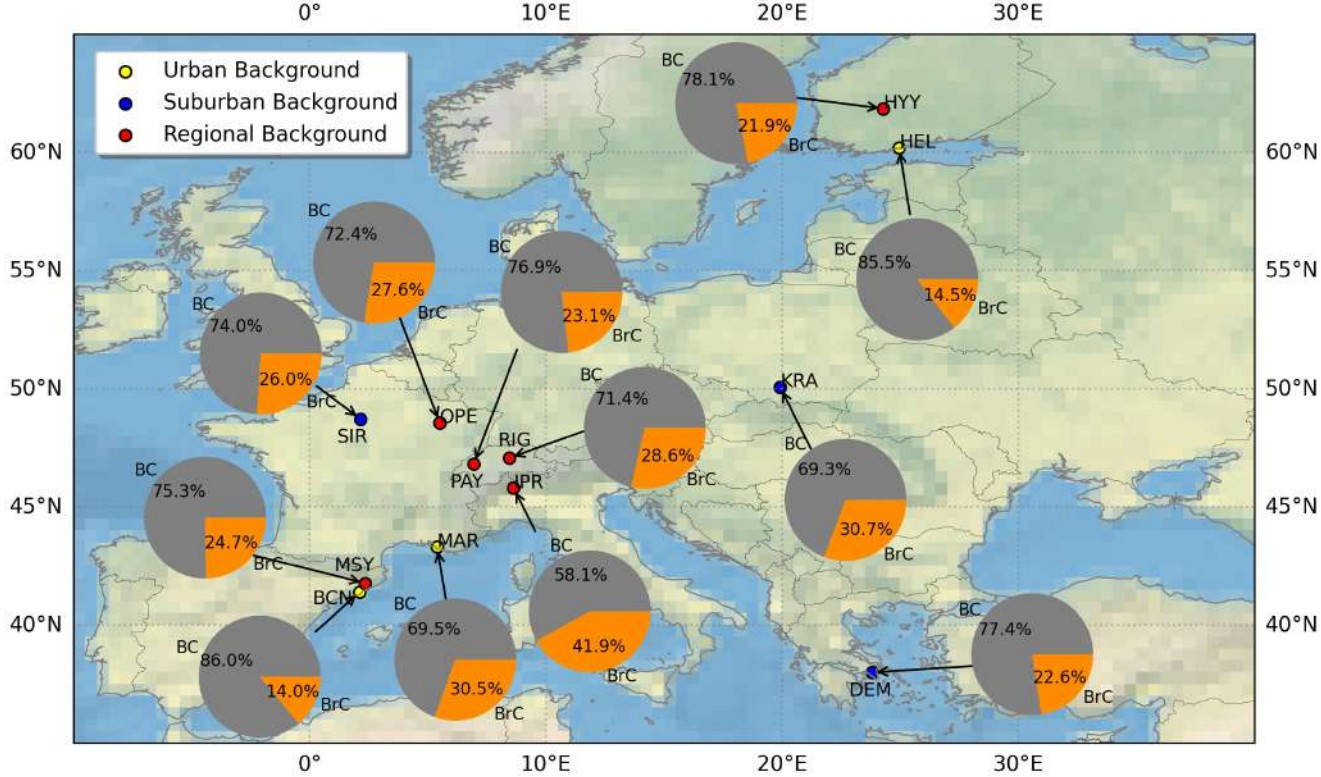

**Figure 1.** Annual average contributions [%] of BrC (orange) and BC (gray) to the total absorption at 370 nm, measured at twelve different monitoring stations across Europe in 2018. The location of each station is indicated by a colored dot, categorized by the station classification: urban background (yellow), suburban background (blue), and regional background (red) environments. The acronyms used for the stations in the figure are explained in the Table 1.

Data from these measurement sites were collected from different infrastructures/projects as EBAS (https://ebas.nilu.no/), RI-URBANS Project (Savadkoohi et al., 2023, https://riurbans.eu/), COLOSSAL COST Action (Chen et al., 2022, https://www.cost.eu/actions/CA16109/), and the FOCI Project (https://www.project-foci.eu/wp/). At some sites (Table 1), OA mass concentrations were directly provided by Aerosol Chemical Speciation Monitor (ACSM) instruments with 30 minute time resolution (Chen et al., 2022), while OC mass concentrations were obtained from the analysis of 24h filters by means of thermal-optical off line technique (SUNSET Instruments) following the EUSAAR II Protocol (Cavalli et al., 2010). Details about ACSM instruments used for OA determination, measurement principle and accuracy of these instruments can be found in Chen et al. (2022), Fröhlich et al. (2013), and Ng et al. (2011). At sites where OC mass concentrations from 24h filters were available (Table 1), specific organic-aerosol-to-organic-carbon (OA/OC) ratios were applied depending on the characteristic of the measurement sites. A factor of 1.4 is traditionally used, although some studies support larger values (e.g., Zhang et al., 2005). Here, we applied OA/OC ratios of 1.8 at urban sites and 2.1 at regional/remote sites. These values agreed with some estimations reported in literature. For example, Minguillón et al. (2011) reported OA/OC ratios of 2.0 and 1.6 for the regional

background, MSY, and urban background BCN sites, respectively. Similarly, Daellenbach et al. (2016) and Favez et al. (2010)

documented OA/OC ratios of 1.84 and 1.8 at two urban background sites in Switzerland and France, respectively.

Aerosol particles light absorption coefficients was derived at 7 different wavelengths (370, 470, 520, 590, 660, 880, and 950 nm) with 1h time resolution using AE33/AE31 Aethalometers (Magee Scientific) instruments. An extensive description of the AE33 and AE31 instruments is provided for example by Drinovec et al. (2015, 2017) and Backman et al. (2017), respectively. Briefly, the Aethalometers measure the attenuation of light by aerosol particles collected onto a fiber filter tape converting the

measured attenuation into absorption.

**Table 1.** Monitoring stations used in this study

| Station | Acronym | Country | Station Type | Variable/ Instrument | Cut-off size Mass/Abs | Time/ Resolution | Latitude | Longitude |
|---|---|---|---|---|---|---|---|---|
| SMEAR II Hyytiala[*] | HYY | Finland | RB | OA/ACSM[a] Abs/AE33[b] | $PM_{10}$/ $PM_{10}$ | 30 min, 1h | 61°51'0.00" N | 24°17'0.00" E |
| Helsinki[*] | HEL | Finland | UB | OA/ACSM[a] Abs/AE33[b] | $PM_1$/ $PM_1$ | 30 min, 1h | 60°11'47.11"N | 24°57'1.31"E |
| Krakow[*] | KRA | Poland | SUB | OA/ACSM[a] Abs/AE33[b] | $PM_1$/ $PM_{2.5}$ | 30 min, 1h | 50°3'56.00" N | 19°54'56.00" E |
| Sirta[***] | SIR | France | SUB | OA/ACSM[a] Abs/AE33[b] | $PM_{2.5}$/ $PM_1$ | 30 min, 1h | 48°42'36.00" N | 2°9'36.00" E |
| Observatoire Perenne de l'Environnement[**] | OPE | France | RB | OC/SUNSET[c] Abs/AE31[d] | $PM_{2.5}$/ $PM_{10}$ | 24h, 1h | 48°33'44.00" N | 5°30'20.00" E |
| Rigi[**] | RIG | Switzerland | RB | OC/SUNSET[c] Abs/AE33[b] | $PM_{2.5}$/ $PM_{2.5}$ | 24h, 1h | 47°4'3.00" N | 8°27'50.00" E |
| Payerne[**] | PAY | Switzerland | RB | OC/SUNSET[c] Abs/AE33[b] | $PM_{2.5}$/ $PM_{10}$ | 24h, 1h | 46°48'47.00" N | 6°56'41.00" E |
| Ispra[**] | IPR | Italy | RB | OC/SUNSET[c] Abs/AE31[d] | $PM_{2.5}$/ $PM_{10}$ | 24h, 1h | 45°49'N | 8°38'E |
| Marseille[*] | MAR | France | UB | OA/ACSM[a] Abs/AE33[b] | $PM_1$/ $PM_{2.5}$ | 30 min, 1h | 43°18'19.1"N | 43°18'19.1"E |
| Montseny[**] | MSY | Spain | RB | OC/SUNSET[c] Abs/AE33[b] | $PM_{10}$/ $PM_{10}$ | 24h, 1h | 41°46'45.63" N | 02°21'28.92" E |
| Barcelona[*] | BCN | Spain | UB | OA/ACSM[a] Abs/AE33[b] | $PM_{10}$/ $PM_{10}$ | 30 min, 1h | 41° 23' 14" N | 2° 06' 56"E |
| Demokritos Athens[*] | DEM | Greece | SUB | OA/ACSM[a] Abs/AE33[b] | $PM_{2.5}$/ $PM_{10}$ | 30 min, 1h | 37°59'24.00" N | 23°49'12.00" E |

[*] Project/Source: RI-URBANS/FOCI/COLOSSAL (Savadkoohi et al., 2023; Chen et al., 2022)

[**] EBAS (https://ebas.nilu.no/)

[***] RI-URBANS/FOCI/EBAS (Savadkoohi et al. (2023); https://ebas.nilu.no/)

[a] OA/ACSM: Organic aerosol/Aerosol Chemical Speciation Monitor

[b] Abs/AE33: Total absorption (370-950 nm)/Aethalometer AE33 model

[c] OC/SUNSET: Organic Carbon/OCEC Carbon Aerosol Analyzer

[d] Abs/AE31: Total absorption (370-950 nm)/Aethalometer AE31 model



### 2.1.1 BrC absorption from aethalometer data

Most of the filter-based absorption techniques that determine the absorption coefficients from the measurements of light passing through an aerosol-laden filter (as the aethalometers), suffer from various systematic errors that need to be corrected. These artifacts include the enhancement of the measured attenuation due to multiple scattering of light by the filter fibers, a further 140 enhancement of light attenuation due to the scattering of aerosols embedded in the filter and a progressive saturation of the instrumental response due to the accumulation of the sample in the filter matrix (e.g., Bond et al., 1999; Weingartner et al., 2003; Moosmüller et al., 2009; Drinovec et al., 2015, 2017; Müller and Fiebig, 2018; Yus-Díez et al., 2021). Thus, absorption data from aethalometer instruments need to be harmonized to take into account for these artifacts. All AE33 data were harmonized as described in Savadkoohi et al. (2023). AE31 data were taken from EBAS Level 2 quality assured/quality checked (QA/QC) 145 dataset and were directly downloaded from the EBAS database (EBAS, https://ebas-data.nilu.no/). These data were processed following the ACTRIS recommendations for the reporting of absorption (Müller and Fiebig, 2018), ensuring the comparability of absorption measurements across the sites by employing harmonized measurement protocols.

The contribution of brown carbon (BrC; $b_{\mathrm{abs},BrC}(\lambda)$) to the total measured absorption ($b_{\mathrm{abs}}(\lambda)$) at different wavelengths from 370 nm to 660 nm was estimated by subtracting the absorption due to BC (BC;$b_{\mathrm{abs},BC}(\lambda)$ to the measured ($b_{\mathrm{abs}}(\lambda)$):

$$b_{\mathrm{abs},BC}(\lambda) = b_{\mathrm{abs}}(880\mathrm{nm}) \cdot \left(\frac{\lambda}{880\mathrm{nm}}\right)^{-AAE_{BC}} \tag{1}$$

$$b_{\mathrm{abs},BrC}(\lambda) = b_{\mathrm{abs}}(\lambda) - b_{\mathrm{abs},BC}(\lambda) \tag{2}$$

where $AAE_{BC}$ is the Absorption Angstrom Exponent (AAE) of BC, which allows for the calculation of $b_{\mathrm{abs},BC}(\lambda)$ (in units of $Mm^{-1}$) from the measurements of $b_{\mathrm{abs},BC}(\lambda)$ at 880 nm assuming that BrC does not absorb at 880 nm (e.g., Qin et al., 2018). The main source of uncertainty in equations 1 and 2 is the AAE assumed for BC. In many studies a value of 155 1 was used (Liakakou et al., 2020; Tian et al., 2023; Cuesta-Mosquera et al., 2023, e.g.,). However, theoretical simulations have shown that the $AAE_{BC}$ can reasonably vary between 0.9 and 1.1 depending on the size and internal mixing of BC particles (Bond et al., 2013; Lu et al., 2015, e.g.,). Here we estimated the site dependent $AAE_{BC}$ as the first percentile of the AAE frequency distribution. The AAE can be calculated from multi-wavelengths (370, 470, 520, 590, 660, 880, and 950 nm) total absorption measurements as the linear fit in a log-log of the total absorption versus the measuring wavelengths. 160 The effect of BrC absorption is to increase the AAE and, consequently, the first percentile of AAE represents conditions where the absorption is dominated by BC. In order to reduce the noise, the 1st percentile at each site was calculated from AAE values obtained from fit with $R^2 > 0.99$ (Tobler et al., 2021). For sites included here, the 1st percentile method provide $AAE_{BC}$ values ranging from 0.928 to 1.088 confirming that this experimental method can provide reasonable estimations of the $AAE_{BC}$. It should be noted that mineral dust particles from North African deserts can absorb at 880 nm even if much less 165 efficiently compared to BC. We assumed that the effect of dust at the surface was present mostly in Mediterranean sites (as BCN, MSY and DEM) due to their proximity to dust source emission (as North African deserts). The measurements possibly affected by dust absorption were removed from the datasets of the above three sites. In the case of BCN and MSY, dusty





days were detected using the methodology that has been officially accepted by the European Commission for reporting on natural contributions to ambient PM levels over Europe (European Commission, 2011). At DEM, dusty days were detected

and removed using the scattering Angstrom exponent (SAE) from in-situ surface nephelometer measurements available in the EBAS database (www.ebas.nilu.no) assuming that SAE values lower than one indicate the presence of dust particles in the atmosphere (e.g. Valenzuela et al. (2015)).

Figure 1 shows the average annual contributions of BC and BrC to the total absorption measured at 370 nm at twelve European stations, identified by color-coded markers indicating their background settings: yellow for urban, blue for suburban,

and red for regional areas. Urban sites, which are BCN, HEL and MAR, report a contribution range from 14% to 30% of BrC, reflecting the significant influence of combustion processes within the cities, with MAR showing a notably high BrC percentage, most probably due to industrial activities and emissions from the port. Suburban stations, including SIR, KRA, and DEM, exhibit BrC proportions from 22% to 30%, reflecting a blend of local urban emissions and regional influences such as biomass burning.

Regional stations, represented by HYY, OPE, RIG, PAY, IPR, and MSY, display BrC levels from 21% to 41%. These percentages indicate a mix of biogenic sources, local emissions, agricultural activities, and trans-boundary pollution affecting the regional atmosphere. IPR stands out with the highest contribution, suggesting significant contribution of low temperature combustion processes (e.g., residential sources).

Overall, although BC typically represents the most absorbing aerosols component at these stations (usually > 70%), it is

noteworthy that BrC could contribute comparably to absorption in some instances.

## 2.2 Model description

### 2.2.1 The MONARCH atmospheric chemistry model

The MONARCH model (Jorba et al., 2012; Badia and Jorba, 2015; Badia et al., 2017; Klose et al., 2021) consists of advanced chemistry and aerosol packages coupled online with the Nonhydrostatic Multiscale Model on the B-grid (NMMB; Janjic et al.,

2001; Janjic and Gall, 2012). The model allows running both global and regional simulations with telescoping nests. Multiple choices of gas- and aerosol chemistry schemes can be selected in the model. Here, we briefly describe the configuration adopted in this work.

The gas-phase chemistry solves the Carbon Bond 2005 chemical mechanism (CB05; Yarwood et al., 2005) extended with chlorine chemistry (Sarwar et al., 2012). The CB05 is well formulated for urban to remote tropospheric conditions, and it uses

photolysis rates computed with the Fast-J scheme (Wild et al., 2000) considering the physics of each model layer (e.g., clouds, absorbers such as ozone). A mass-based aerosol module describes the life cycle of dust, sea salt, BC, OA (both primary and secondary), sulfate, ammonium and nitrate aerosol components (Spada, 2015). A sectional approach is used for dust and sea salt, while the other aerosol species are represented by a fine mode, except nitrate which is extended with a coarse mode to consider the condensation of nitric acid on coarse particles. Sulfate production considers the gas-phase oxidation of both sulfur

dioxide ($SO_2$) and dimethyl sulfide, and the aqueous chemistry of $SO_2$. The heterogeneous hydrolysis of $N_2O_5$ contributes to





the production of nitric acid using the parameterization of Riemer et al. (2003). A thermodynamic equilibrium model (Metzger et al., 2002) solves the partitioning of semivolatile inorganic aerosol components in the fine mode, and an irreversible uptake reaction accounts for the production of coarse nitrate in dust and sea salt (Hanisch and Crowley, 2001; Tolocka et al., 2004). Different meteorology-driven emissions are computed online in MONARCH (i.e., mineral dust, sea salt, and biogenic gas
species). The mineral dust scheme of the model is described in detail in Pérez et al. (2011) and Klose et al. (2021). Sea salt emissions are calculated following the source function of Jaeglé et al. (2011) as described in Spada et al. (2013), while biogenic Non-Methane Volatile Organic Compounds (NMVOC) and soil NO emissions are estimated with the Model of Emissions of Gases and Aerosols from Nature (MEGAN) v2.04 model (Guenther et al., 2006).

Black carbon is represented in MONARCH following Chin et al. (2002). Two primary hydrophobic/hydrophilic modes are
defined with an aging process converting mass from the hydrophobic to the hydrophilic mode with a lifetime of 1.2 days. Primary emissions are assumed to be emitted as 80% hydrophobic.

The simple scheme proposed in Pai et al. (2020) is adopted to model OA. It is computationally efficient and reproduces well the organic mass assuming fixed SOA yields adjusted to match results from the more complex volatility-based scheme approach. Here, we briefly describe the scheme. Primary emissions are emitted as 50% hydrophobic species with an OA/OC ratio
of 1.4, while the hydrophilic oxygenated component assumes an OA/OC ratio of 2.1. Similarly to BC, an atmospheric aging of hydrophobic to hydrophilic primary species is simulated with a conversion lifetime of 1.15 days. In our implementation, no marine primary organic aerosol is considered. The scheme includes sources of SOA precursors from biogenic, pyrogenic, and anthropogenic origin with fixed SOA yields. Biogenic sources of SOA uses a 3% yield for isoprene and 10% yield for both monoterpenes and sesquiterpenes. A 50% of biogenic SOA is emitted directly to account for the near-field formation of SOA.
On the other hand, precursors from combustion emissions are scaled from CO emissions as a proxy, of which 1.3% come from fires and biofuels (combustion sources) and 6.9% from fossil fuels. The gas-phase SOA products converts to the aerosol phase based on a first-order rate constant with a lifetime of 1 day.

### 2.2.2 Optical properties

We use an offline optical package (Obiso, 2018) to calculate the absorption by OA using the mass concentration simulated
by the model. The package allows calculating intensive optical properties of a size-distributed particle ensemble, including the absorption efficiency ($Q_a$) as the difference between extinction and scattering efficiencies (Mishchenko et al., 2002). The required input physical properties of the aerosols are the size distribution, the complex refractive index, the particle shape and the hygroscopicity. Our package only uses the log-normal size distribution, that is defined by two parameters: the geometric radius ($r_g$) and the standard deviation ($\sigma_g$). The real ($n$) and imaginary ($k$) parts of the complex refractive index determine
the scattering and absorption properties of the particles, respectively, and primarily depend on their internal composition. The spherical shape is assumed by default in the package while the water uptake of the hydrophilic modes is taken into account through a grid of hygroscopic growth factors ($\alpha$) defined for specific values of relative humidity ($RH$). The assumption of external mixture is adopted in the calculation of the absorption by OA.



The package is built on a data set of monodisperse single-wavelength optical properties, pre-calculated using the Mie-theory
code by Mishchenko et al. (2002), whose structure allows computational efficiency while preserving application flexibility.
In general, the optical properties of a single particle depend on the ratio of its size to the incident wavelength, rather than on
those two quantities separately (Mishchenko et al., 2002). For this reason, the data set is calculated on a grid of size parameters
($x = 2\pi r/\lambda$, where $r$ and $\lambda$ are the particle radius and the incident wavelength, respectively) ranging from $0.011$ to $\sim 1000$.
Following Gasteiger and Wiegner (2018), we apply an increment of $1\%$ to each grid value $x_i$ to obtain the next one; moreover,
we store the optical properties integrated in very narrow size bins centered in each $x_i$ and ranging from $x_i/\sqrt{1.01}$ to $x_i\sqrt{1.01}$.
The entire set of size parameters is then considered for a grid of real indexes (ranging from $1.3$ to $2$, with a step of $0.05$) and
imaginary indexes (from $0$ to $1$, with varying resolution across different orders of magnitudes).

The physical properties of OA used in this work, that align with the model aerosol representation, are presented in Table
2. Size distribution and standard refractive indexes are taken from the Optical Properties of Aerosols and Clouds (OPAC)
database (Hess et al., 1998). The hygroscopic growth factors for seven prescribed $RH$ values are from Chin et al. (2002).
Once defined the working wavelength of 370 nm, which allows mapping the size parameters onto particle radii, the needed
pre-calculated optical efficiencies are integrated over the input size distribution. Then, the actual extinction and scattering
efficiencies corresponding to the input refractive index are obtained by means of bilinear interpolation from the integrated
values at the four closest gridded refractive indexes. The water uptake affects both the input size distribution and refractive
index of the hydrophilic modes of OA. Once read a specific $RH$ level calculated by the model, the corresponding hygroscopic
growth factor is set by linearly interpolating between the closest gridded values and subsequently applied to the geometric
radius of the size distribution (as well as to the extremes of the size integration): $r_{g,w} = \alpha r_g$. The refractive index of the wet
particles is obtained as the volume-weighted mean of the refractive index of the dry particles and that of water (the latter taken
from Segelstein, 1981).

Once obtained the size-integrated absorption efficiency, the absorption coefficient of OA is calculated as follows:

$$b_a = \frac{3Q_{a,w}\alpha^3}{4\rho r_{e,w}}M \tag{3}$$

where $Q_{a,w}$ is the wet absorption efficiency, $\alpha^3$ is the wet-to-dry mean volume ratio, $\rho$ is the mass density of the dry
particles (Table 2), $r_{e,w}$ is the wet effective radius (defined as the projected-surface-weighted mean radius) and $M$ is the mass
concentration of OA from the model. Note that the wet quantities only refer to the hydrophilic OA modes and tend towards the
corresponding dry quantities for the hydrophobic mode (for which $\alpha = 1$).




**Table 2.** Physical properties of the organic aerosol species implemented in the MONARCH model and used in this work for optical calculations: geometric radius ($r_\mathrm{g}$), standard deviation ($\sigma_\mathrm{g}$) and effective radius ($r_\mathrm{e}$) of the size distribution, real ($n$) and imaginary ($k$) refractive indexes, mass density ($\rho$) and hygroscopic growth factor ($\alpha$). In the second column, *phob* stands for "hydrophobic mode" and *phil* for "hydrophilic mode". The range extremes used for size integration ($r_1$–$r_2$) are reported within parentheses close to the corresponding $r_\mathrm{e}$ values. The seven values for $\alpha$ apply to the hydrophilic modes and are relative to $RH$ levels of 0%, 50%, 70%, 80%, 90%, 95% and 99%. The refractive indexes reported are relative to a wavelength of 550 nm.

| Parameters | Modes | Organic Aerosol | |
|---|---|---|---|
| $r_\mathrm{g}$ ($\mu$m) | *phob-phil* | $2.12 \cdot 10^{-2}$ | |
| $\sigma_\mathrm{g}$ | *phob-phil* | 2.2 | |
| $r_\mathrm{e}$ ($\mu$m) | *phob-phil* | $1.003 \cdot 10^{-1}$ | (0.005–20) |
| $n$ | *phob-phil* | 1.53 | |
| $k$ | *phob-phil* | $6.0 \cdot 10^{-3}$ to $> 0.1$ | |
| $\rho$ ($\mathrm{g\,cm^{-3}}$) | *phob-phil* | 1.8 | |
| $\alpha$ | *phob* | 1.0 | |
| | *phil* | (1.0, 1.2, 1.4, 1.5, 1.6, 1.8, 2.2) | |

### 2.2.3 Emissions

The High-Elective Resolution Modelling Emission System version 3 (HERMESv3; Guevara et al., 2019, 2020) is used to provide anthropogenic, biomass burning, and ocean emissions to be used as input in the MONARCH model. In this study, we employ the global-regional module (HERMESv3_GR; Guevara et al., 2019), which allows users to flexibly combine gridded global and regional emission inventories. In addition, this module facilitates the application of country-specific scaling factors and masks. HERMESv3 disaggregates the original datasets both spatially and temporally, and applies user-defined speciated emissions.

In this study, we used the European-scale emission inventory CAMS-REG-AP_v4.2 (Kuenen et al., 2022), developed under the Copernicus Atmosphere Monitoring Service (CAMS). Within this framework, the CAMS-REG-AP_v4.2_REF2 represents a specific dataset or version of the CAMS-REG-AP_v4.2 inventory, which adopts a harmonized approach for consistently including the condensable fraction for the residential wood combustion (RWC) particulate matter emissions. This is important to address the previously identified inconsistencies across European inventories, as highlighted by Denier Van Der Gon et al. (2015), which mainly stem from variable emission factors used by different countries accounting or not for condensables impacting modeling results (e.g., Navarro-Barboza et al., 2024). To accurately quantify biomass burning (BB) emissions, our study utilized data from the Global Fire Assimilation System version 1.2 (GFASv1.2) analysis. This dataset provides detailed emission fluxes derived from satellite information for various sources such as forest, grassland, and agricultural waste fires (Kaiser et al., 2012). Additionally, for oceanic dimethyl sulfide (DMS) emissions, we relied on the CAMS Global Ocean dataset (Lana et al., 2011; Granier et al., 2019).





## 2.3 Model simulations

In this study, we used the MONARCH model with a domain that covers the European continent and part of North Africa at a horizontal resolution of $\sim 20$ km, as shown in Figure 2. Perturbation runs (commonly know as the brute force method) were conducted to apportion the contribution from fires, traffic, shipping, residential, and other sources. Biomass burning emissions derived from the GFASv1.2 product are tagged (GFAS) as one of the main contributors to OA absorption. Traffic emissions (TRAF) categorized under sectors GNFR_F1, F2, F3, and F4 account for exhaust and non-exhaust emissions of gasoline,

diesel and liquefied petroleum gas vehicles. Shipping emissions (SHIP) are derived from GNFR_D sector. The emissions from commercial, institutional and residential sources (RESI) consider a wide range of sources related to buildings and facilities and are categorized under sector GNFR_C. RESI includes activities of combustion in different types of devices, including boilers, turbines, engines, and chimneys, for different fuel types (i.e., natural gas, wood, fuel oil, LPG, coal). In this sector, only combustion activities related to space heating, cooking, and water heating are included (cleaning activities are not considered).

Furthermore, the rest of sources are tagged together as other sources (OTHR) including public power, industry, fugitives, solvents, aviation, offroad, waste, agriculture, and biogenic emissions.

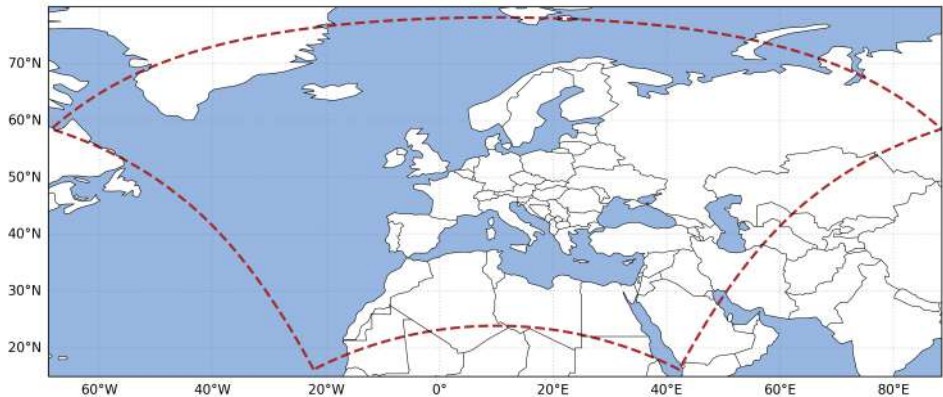

**Figure 2.** Model domain at $\sim 20$ km of horizontal resolution.

The model ran with 24 vertical layers and a top of the atmosphere set at 50 hPa. Meteorology initial and boundary conditions were obtained from the ECMWF global model at $0.125°$, and the chemical boundary conditions from the CAMS global system at $0.45°$ (Flemming et al., 2015). Emissions were processed as described in Section 2.2.3. The atmospheric meteorological

variables are initialized every 24h to keep the modeled circulation close to observations, while the chemistry initial conditions are those prognostically estimated by MONARCH (i.e., every day the model uses as an initial state for these variables their modeled value at 24 UTC of the day before). A spin-up period of 15 days is used to derive the chemistry initial conditions for 2018.

For efficient execution of the MONARCH modeling chain, we utilized the autosubmit workflow manager, a tool proven

effective in such complex modeling simulations (Manubens-Gil et al., 2016).



## 2.4 Off-line Refractive index optimization

Since $k$ is a highly uncertain parameter with a strong dependence on the sources of OA, we employed a method to derive an optimized $k$ for each OA component (aggregating both primary and secondary contribution) that combines the results of the perturbation runs, which provides source apportionment of OA mass, and observations described in Section 2.1.

Based on the aerosol mass calculated by the model, we use the offline optical tool described in Section 2.2.2 to derive $k$ values that minimize the error with the absorption measured at the 12 monitoring stations across Europe (see Figure 1). The optimized $k$ values are derived using the SLSQP algorithm (Sequential Least Squares Programming), which is particularly well-suited for nonlinear objective functions and constraints to minimize a cost function. This process handles the task of minimizing the error between the modeled and the observed absorption. The cost function calculated here evaluates this error at each

time step, guided by predefined boundary conditions for $k$ values obtained from Saleh (2020). These conditions categorize OA absorption into four optical classes: very weakly absorptive OA (VW-OA), weakly absorptive OA (W-OA), moderately absorptive OA (M-OA), and strongly absorptive OA (S-OA). The relation between OA absorbing categories and OA sources is described in Section 2.5.

     For the computation of $k$ at 370 nm within the optimization module, we adopt the equation described in Saleh (2020), which

describes the wavelength dependency of $k$ for BrC:

$$k(\lambda) = k_{550} \times \left( \frac{550}{\lambda} \right)^{w} \tag{4}$$

where $k_{550}$ represents the imaginary refractive index ($k$) at 550 nm, $\lambda$ is the wavelength in nm, and $w$ denotes the wavelength dependence. The values of $w$ and $k_{550}$ for each OA class (VW-OA, W-OA, M-OA, and S-OA) were determined based on Table 1 in Saleh (2020).

Regarding the constraints used in the SLSQP algorithm, we have applied the condition which is based on extensive experimental and modeling results e.g., (Andreae et al., 1998; Bond et al., 2013; Laskin et al., 2015; Saleh, 2020), suggesting that OA from fires exhibit the highest $k$ values.

     Figure 3 illustrates the steps of the optimization process. The procedure starts with the tagging of OA per source derived from MONARCH runs, followed by the application of a priori $k$ values imposed based on Saleh (2020). Note that Saleh (2020)

categories for OA absorption were derived from 20 chamber experiments that may not be completely representative of ambient conditions in the field. The subsequent steps include calculating absorption and determining $k$ values for each source with the SLSQP algorithm. The process concludes with the resulting optimization of $k$ values at 370 nm for each component and the calculation of the final OA absorption.



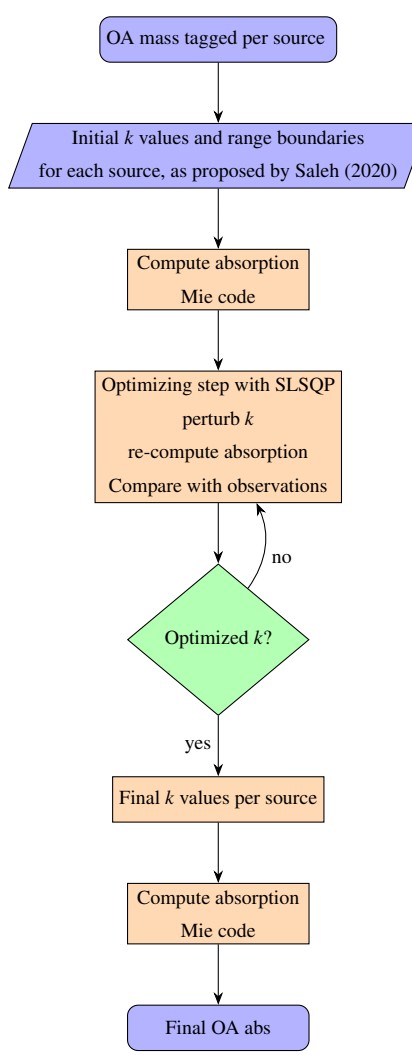

**Figure 3.** Steps to derive optimized imaginary refractive index $k$ for total or tagged OA.

The resulting $k$ are representative of OA in environmental conditions, which may include interactions with other species,
providing an empirical range of $k$ values that account for various OA sources and potential interference.

The optical calculation relies on the external mixing assumption. Attempting to constrain $k$ of OA by assuming internal mixing or core-shell configurations with other species would be complex and unlikely to yield results with less uncertainty. The internal mixing approach would introduce additional modeling uncertainties, such as mixing rules, refractive indices of other species, and OA fractions relative to these species. Therefore, assuming externally mixed OA is considered a simpler
approximation and more suited to constrain source specific contributions, despite its inherent limitations.



## 2.5 Scenario-Based Approach for *k* Optimization

To investigate the absorption characteristics of different OA sources we have defined six distinct optimization cases, labeled Case 1 through Case 5, as detailed in Table 3. In particular, each case provides the boundaries for the optimization method described above that are used to find the optimal *k* value for a specific OA source. Note that the boundaries limit the range where the SLSQP algorithm employed searches for an optimized solution.

Cases 1 to 4 were confined within specific *k* range boundaries at 370 nm, as recommended by Saleh (2020). These recommended ranges, originally specified at 550 nm based on chamber experiments, were adapted to 370 nm using the wavelength dependence (*w* values) provided in Table 1 of Saleh (2020). It is important to note that while the *k* ranges at 370 nm are broader and may overlap between categories, this is attributed to the greater wavelength dependence observed. Case 5 was designed to derive a *k* value without any predefined boundaries or constraints and applied to total OA mass (not per emission source), providing a flexible approach to understanding OA absorption.

To enhance our methodology, we introduced two additional strategies to optimize *k*. The first approach involves optimizing *k* individually at each monitoring station, allowing for a tailored assessment that accounts for local variations in the properties of OA. The second strategy consolidates data from all stations to derive a single unified *k* value, providing a broader and more generalized perspective on OA absorption characteristics. Both methods were applied separately for specific sources of OA and for the total OA observed, enabling a comprehensive analysis of the influence of source-specific and aggregate OA contributions on absorption properties. This dual approach ensures a robust optimization process that accommodates both localized and generalized environmental conditions.

As detailed in Table 3, Case 1 categorizes OA from fires (GFAS), residential (RESI), and shipping sources (SHIP) as weakly absorbing, with *k* values ranging from 0.0049 to 0.1604. In contrast, traffic (TRAF) and other sources (OTHR) are considered very weakly absorbing, with *k* values from 0.0011 to 0.0354. Case 2 adjusts the absorption levels for OA from GFAS, RESI and SHIP to moderately absorbing, with *k* values from 0.0181 to 0.4883, while maintaining the same very weak absorption for TRAF and OTHR sources as observed in Case 1. In Case 3, OA from GFAS are assigned strong absorption properties with *k* values from 0.1219 to 0.6887, whereas RESI and SHIP are treated as having moderate absorption, similar to the *k* values in Case 2, and TRAF and OTHR sources continue to be categorized as very weak absorbers. Case 4 adjusts the absorption categorization, treating fires with moderate absorption, while TRAF, SHIP, and RESI are considered weakly absorbing, and OTHR remains very weakly absorbing. Case 5 involves calculating the optimized *k* by integrating all OA components (ALL) without predefined boundaries, facilitating a comprehensive exploration of *k* values. Note that contribution from both primary and secondary aerosols is accounted within each OA categorization.





**Table 3.** Scenarios and ranges of *k* used in the optimization process.

| Case | Source | Absorption Category | *k* range boundaries at 370 nm |
|------|--------|---------------------|-------------------------------|
| Case 1 | Fires, Residential, Shipping | Weakly | 0.0049 to 0.1604 |
| | Traffic, Others | Very Weakly | 0.0011 to 0.0354 |
| Case 2 | Fires, Residential, Shipping | Moderately | 0.0181 to 0.4883 |
| | Traffic, Others | Very Weakly | 0.0011 to 0.0354 |
| Case 3 | Fires | Strongly | 0.1219 to 0.6887 |
| | Residential, Shipping | Moderately | 0.0181 to 0.4883 |
| | Traffic, Others | Very Weakly | 0.0011 to 0.0354 |
| Case 4 | Fires | Moderately | 0.0181 to 0.4883 |
| | Traffic, Shipping, Residential | Weakly | 0.0049 to 0.1604 |
| | Others | Very Weakly | 0.0011 to 0.0354 |
| Case 5 | Total OA | N/A | N/A |

## 3 Results

### 3.1 OA mass and source contribution

Our study focuses on understanding the light absorption properties of OA across different European environments. Since the light absorption of OA ($b_a$) is intrinsically linked to its mass concentration, a first step is to evaluate the accuracy and reliability of our model in simulating the mass concentrations. In this section, we examine the model's ability to describe the variability

of OA mass concentration at the 12 monitoring stations presented in Table 1.

Figure 4 shows the time series of the measured and modeled OA mass concentrations for 2018 (note the varying y-axis scales in different panels). For a detailed comparison of modeled versus observed OA concentrations across the same stations, refer to the scatter plots provided in the supplementary material (see Figure S2). Observational data (OBS), depicted by red dots, show the actual OA concentrations measurements at each station. These values provide a baseline against which the model's

performance can be evaluated. The modeled concentrations of OA, derived from the source-tagged simulation, are represented by filled colors, where each color shows the contribution from different emission sources. Specifically, SHIP is marked in purple, RESI emissions in light blue, GFAS in orange, TRAF in black, and OTHR in brown (only primary contribution shown). Additionally, Secondary Organic Aerosols (SOA) contribution is depicted in green.

As reported in Figure 4 and Table 4, overall a good agreement was observed between measured and modeled OA concentra-

tions in terms of statistics metrics (see below). This indicates the model ability to capture the pronounced seasonal trends and the peak concentrations at the majority of the monitoring stations studied. For example, one episode of increased concentration



in Marselle site (MAR) in February, when the observed concentrations reached 9.1 $\mu gm^{-3}$, was closely approximated by the model which reported 7.4 $\mu gm^{-3}$. Similarly, a significant event in Ispra (IPR) towards the end of December with observed OA concentration of 27.8 $\mu gm^{-3}$ was closely simulated by the model with a value of 27.4 $\mu gm^{-3}$.

Notably, the residential component (in blue) emerges as the predominant source across all monitoring sites. A consistent seasonal pattern is evident, particularly during the colder months, which is likely associated with the increase in residential heating during these months. It is important to note that Navarro-Barboza et al. (2024) highlighted notable inaccuracies in the carbonaceous aerosol emissions attributed to the residential sector within the CAMS_REGv4.2 emission inventory for some stations in the western Mediterranean (namely Barcelona (BCN) and Montseny (MSY) sites), suggesting an overestimation of

its contribution which was particularly relevant during winter. This issue is further detailed in the Supplemental Material Figure S1 where OA concentrations for BCN and MSY stations (January and July presented) were simulated using a local bottom-up emission inventory for Spain (Guevara et al., 2020) showing that traffic emissions appear more significant compared to residential emissions in those sites. This represents a limitation of the dataset utilized in this study and underscores the need for ongoing enhancements of continental-scale emission inventories like CAMS_REGv4.2, which are currently among the best

resources for modeling studies in Europe.

     The second most significant contributor to the mass of OA is attributed to secondary organic aerosols (SOA, highlighted in green), which are especially important during the summer months. SOA shows important spatial and temporal variability. For instance, SOA mass formation becomes more significant during warm periods, especially as observed in HYY, where the model tends to underestimate the high observed OA concentrations in July associated with high SOA mass formation

(Heikkinen et al., 2021; Yli-Juuti et al., 2021). Similarly, the model underestimates the high OA concentrations observed in MSY in summer that were also driven by the high formation of SOA (e.g., In't Veld et al., 2021). The SOA yields utilized in this model simulation for biogenic precursors, as derived from Pai et al. (2020), might result in limited SOA production and could account for this negative bias.

     Other contributions (OTHR) increase during the warm months in most stations, except KRA and IPR. The increase in activity

could be related to the increased agricultural practices prevalent during the warmer months (e.g., agricultural waste burning). The less pronounced trend in KRA and IPR could be due to different agricultural practices, reduced agricultural activity, or more effective waste management strategies during these months. The emissions of SHIP contribute significantly to stations near ports, such as BCN, DEM, and MAR. Traffic-related emissions remain fairly constant and low across all sites; according to Chen et al. (2022), however, this component should not only be more pronounced at urban stations such as BCN, MAR and

HEL, but the contribution should not be neglected at non-urban stations either. The underestimation of TRAF emissions may indicate a need for refinement in the emission inventory as discussed above and illustrated in the Supplementary Material for BCN site (Figure S1).

     Specific events, such as forest fires or high pollution episodes, are reflected in pronounced peaks in both modeled and observed concentrations, highlighting the model's ability to respond to such episodes. Notably, a significant peak is observed

towards the end of July in the MAR and DEM stations, potentially linked to specific forest fire event, as depicted by the model fires contribution (orange color in Fig. 4). This is corroborated by observations at DEM, although, no observational data




is available for MAR for that episode to validate the output of the model. However, the model does not appear to effectively capture high-pollution events in KRA during the winter, where peaks exceeding 60 $\mu g/m^3$ were monitored. KRA is recognized as a major pollution hotspot in Europe (e.g., Casotto et al., 2023), largely due to the extensive use of coal and wood for energy

production and residential heating. This likely accounts for the elevated OA levels monitored in KRA during the winter of 2018, given that the prohibition on solid fuels (coal and wood) in boilers, stoves, or fireplaces was only implemented after September 2019. Additionally, KRA's location within a basin with poor ventilation makes it susceptible to air pollution buildup (Sekula et al., 2022). These factors indicate potential shortcomings in the emissions data and the model's ability to accurately simulate strong inversion events.

**Figure 4.** Time series of observed and modeled daily mean OA mass concentrations [$\mu g m^{-3}$] for the twelve European monitoring stations. Contributions from various emission sources to primary organic aerosol including Shipping (SHIP), Residential (RESI), Fires (GFAS), and Other (OTHR) are presented. Secondary Organic Aerosol (SOA) is shown as a separate component. Observed data points are indicated with red points. Gray background indicates that the bias is within an acceptable range ($\pm 1.5$ $\mu g m^{-3}$). Y-axis scales differ across panels.





Our analysis is further complemented by a statistical evaluation of OA mass concentrations using five metrics: Normalized Mean Bias (NMB), Normalized Mean Error (NME), Pearson Correlation Coefficient (r), Fraction of Predictions within a Factor of Two (FAC2) of observations, and Fractional Bias (FB). We followed the performance assessment approach recommended by Emery et al. (2017) for photochemical models, particularly for pollutants such as Elemental Carbon (EC) and OC, focusing on metrics including FB, NME, and NMB. These recommendations are based on the "goal" and "criteria" benchmarks proposed

by Boylan and Russell (2006), where the "goal" signifies the peak performance expected from a model, and the "criteria" represents a level of performance that should be achievable by most models. The metrics are defined in supplementary material Table S1.

    Table 4 summarizes the statistical evaluation for the twelve stations (DJF December-January-February; MAM March-April-May; JJA June-July-August; SON September-October-November). Overall, the "criteria" is achieved in most of the stations

and seasons, while some sites do not meet the "goal". We observe that the model exhibits its strongest performance during the spring season (MAM), with many stations meeting the "goal" and "criteria" benchmarks. Conversely, the summer (JJA) and winter (DJF) seasons show more variation in the model's ability to accurately reproduce the observed OA concentrations.

    During winter (DJF), for instance, the HYY station shows a strong correlation ($r = 0.94$) but a low FAC2 value ($FAC2 = 23.53$). This discrepancy suggests a tendency to overestimate concentrations, which could be attributed to an overrepresentation

of residential emissions in the model during this season. In contrast, during spring (MAM), the HEL station achieves a high FAC2 value (93.59%) along with a good correlation ($r = 0.72$), which might be attributed to a balanced influence from various sectors, particularly as fire emissions become more significant during this period (Urbieta et al., 2015; Turco et al., 2017). Most stations have better performance in MAM compared to other seasons, with many achieving their "goal" achievements and benchmarks in NME and NMB, such as SIR with low fractional bias and normalized errors. Notably, a drop in correlation

is observed during summer (JJA) and autumn (SON) at some stations (e.g., KRA in SON with $r = 0.34$), but several maintain high FAC2 values, like PAY in JJA (100%) and RIG in SON (100%). This suggests that the discrepancy could stem from challenges in accurately modeling biogenic SOA contributions during the warmer months. The gray highlighting for stations like SIR and OPE in these seasons indicates success in achieving targeted goals for NME and NMB, despite some challenges in FB. Unfortunately, data from stations such as, HEL and MAR is unavailable for the JJA and SON seasons.




**Table 4.** Seasonal statistical evaluation of OA concentrations. Goal achievement (Gray) and criteria benchmarks (Bold) based on Emery et al. (2017). NMB: Normalized Mean Bias, NME: Normalized Mean Error, and FAC2 expressed as percentages.

| Metric | r | FAC2 | FB | NME | NMB | r | FAC2 | FB | NME | NMB | r | FAC2 | FB | NME | NMB | r | FAC2 | FB | NME | NMB |
|---|---|---|---|---|---|---|---|---|---|---|---|---|---|---|---|---|---|---|---|---|
| Season | DJF | DJF | DJF | DJF | DJF | MAM | MAM | MAM | MAM | MAM | JJA | JJA | JJA | JJA | JJA | SON | SON | SON | SON | SON |
| STN | | | | | | | | | | | | | | | | | | | | |
| HYY | 0.94 | 23.53 | 84.43 | 146.11 | 146.11 | 0.64 | **79.17** | **-9.02** | **45.37** | **-8.63** | 0.84 | 73.56 | -52.56 | 43.73 | -41.62 | 0.83 | 72.13 | **-3.78** | 39.10 | -3.71 |
| HEL | 0.65 | **66.10** | **45.89** | 70.61 | 59.55 | 0.72 | **93.59** | **-3.64** | **33.46** | **-3.58** | nan | nan | nan | nan | nan | nan | nan | nan | nan | nan |
| KRA | 0.53 | **73.33** | **-26.97** | **44.80** | **-23.77** | 0.67 | **90.48** | **-20.53** | **32.72** | -18.62 | 0.55 | 65.43 | -55.38 | 44.73 | -43.37 | 0.34 | 74.63 | -32.54 | 41.06 | -27.99 |
| SIR | 0.61 | **61.84** | **28.93** | 58.73 | 33.82 | 0.55 | **79.27** | **-3.06** | **37.00** | **-3.02** | 0.63 | **87.78** | **-12.93** | 38.48 | **-12.14** | 0.36 | 66.25 | 32.90 | 64.81 | 39.37 |
| OPE | 0.80 | **70.59** | **-4.48** | 45.92 | **-4.38** | 0.85 | 68.42 | -52.85 | 43.28 | -41.81 | 0.57 | 66.67 | -51.30 | 43.48 | -40.83 | 0.78 | **85.00** | 29.82 | 49.58 | 35.05 |
| RIG | 0.94 | 40.00 | **10.77** | 53.95 | 11.38 | -0.06 | **71.43** | **-14.61** | 50.09 | **-13.62** | 0.77 | **75.00** | **-24.56** | 32.22 | -21.88 | 0.61 | **100.00** | **3.16** | 26.98 | **3.21** |
| PAY | 0.84 | **76.47** | **-12.39** | **32.32** | **-11.67** | 0.09 | **85.71** | **-13.43** | **28.80** | **-12.58** | 0.86 | **100.00** | -32.95 | 33.78 | -28.29 | 0.51 | **87.50** | -31.94 | 29.23 | -27.54 |
| IPR | 0.62 | **61.54** | -50.53 | **43.89** | -40.34 | 0.33 | 49.32 | **17.31** | 70.19 | **18.95** | 0.26 | **75.28** | **17.77** | 45.24 | **19.50** | 0.52 | **71.43** | **21.84** | 52.85 | 24.51 |
| MAR | 0.61 | **80.77** | **-29.05** | **37.95** | -25.36 | 0.49 | **77.78** | 38.48 | 63.81 | 47.64 | 0.48 | **96.00** | **15.29** | **33.37** | **16.56** | nan | nan | nan | nan | nan |
| MSY | 0.66 | 14.29 | 95.00 | 180.93 | 180.93 | 0.39 | **57.14** | 54.86 | 86.62 | 75.59 | 0.63 | **86.96** | -31.59 | 29.38 | -27.28 | 0.52 | **70.00** | 48.39 | 69.02 | 63.84 |
| BCN | 0.58 | 41.82 | 70.13 | 112.18 | 108.01 | 0.62 | **80.00** | **28.34** | **44.92** | 33.02 | 0.80 | **86.96** | -37.33 | **32.21** | -31.46 | 0.37 | **75.00** | **13.24** | 45.74 | **14.17** |
| DEM | 0.47 | **83.33** | **-0.40** | **36.28** | **-0.40** | 0.41 | **86.21** | -38.31 | **35.59** | -32.15 | 0.38 | 72.22 | -50.43 | 42.66 | -40.27 | 0.58 | **83.02** | -32.00 | **30.10** | -27.59 |

## 3.2 Imaginary refractive index (*k*) optimization

The optimization of the imaginary refractive index (*k*) at 370 nm for OA across the twelve measurement stations presents challenges, particularly due to limitations in data coverage. The data points used for the optimization process were selected based on their consistency with observed OA concentrations. However, the availability of data varied significantly between stations, imposing limitations on the optimization process. For instance, the HEL station provided data only from January to May, and the MAR station had a similarly limited data range. The HYY station data coverage spanned from the end of February to the end of October. Other stations, such as OPE, PAY, RIG, and MSY, contributed relatively few data points throughout the year. This gaps in data coverage could potentially impact the representativeness of the *k* optimization results.

The selection criteria for the data points to be used in the optimization process included a bias threshold, where only days with bias within $\pm 1.5$ $\mu g m^{-3}$ of the actual OA measurements were considered (identified in gray background in Figure 4). This approach aimed to ensure the reliability of the optimization results despite limitations in data availability.

Once the OA measurements had been selected as described above, the aim of the *k* optimization procedure applied was to determine the most suitable *k* for the different sources considered that would optimize the comparison between the simulated and observed OA absorption coefficient.

Table 5 presents the mean and standard deviation (std) of the optimization results for *k* across stations, classified by representative environments (Regional - REG, Suburban - SUB, Urban - URB; see Table 1), and optimization cases 1 through 4 (Table 3). Each case corresponding to unique assumptions discussed in Section 2.5. The mean *k* and std were analyzed for the five sources of study: GFAS, RESI, SHIP, TRAF, and OTHR. The row labeled "ALL" indicates the average values for all stations in Case 4.

By comparing Table 3 and Table 5, the results of the performed *k* optimization process can be summarized as follow. Overall, for all settings and Cases (1-4), and also for ALL, the highest *k* was obtained for GFAS compared to the other sources with



the highest *k* observed for URB sites, whereas similar k within the std were obtained at REG and SUB sites. In each setting, small differences were observed between Case 1 (GFAS weakly) and Cases 2 and 4 (GFAS moderately) indicating a robust estimation of GFAS *k* in-between weakly and moderately absorbing OA particles. Interestingly, in Case 3 (GFAS strongly) *k* reached the lowest allowed limit (cf. Table 3) with zero std (cf. Table 5) suggesting that GFAS OA particles cannot be treated
as strongly absorbing.

RESI OA particles were considered as weakly absorbing (Cases 1 and 4) and moderately absorbing (Cases 2 and 3). Similarly to GFAS, higher RESI *k* was obtained for URB sites even if for this source the obtained *k* were comparable, within the std, among the three considered settings. This similarity could be associated to the fact that RESI emissions are mostly local, thus reducing the differences among the station settings. Moreover, for each setting small differences were observed for RESI
*k* among the 4 considered Cases suggesting a robust estimation of the RESI *k* that lied in-between weakly and moderately absorbing OA particles. Note that for both GFAS and RESI the optimization process provided *k* values closer to the upper limit of the weakly category rather than the upper limit of the category moderately.

Similarly to RESI, SHIP emissions were considered as weakly absorbing in Cases 1 and 4 and moderately absorbing in Cases 2 and 3. Overall, the optimization process provided rather similar *k* among the four Cases at URB sites with *k* values
that were higher compared to REG and SUB. At URB sites, the SHIP *k* lied in-between the categories weakly and moderately with *k* values that, as for GFAS and RESI, were closer to the lower limit of the category moderately. At SUB sites, the SHIP *k* reached the lowest limit allowed for the category weak (0.005; Cases 1 and 4) and moderately (0.02; Cases 2 and 3) suggesting very low *k* of SHIP for this setting. At REG sites, SHIP *k* was higher than at SUB sites with low variability among Cases 1, 2 and 4 and higher values and much higher std for Case 3. Also for REG sites the optimization process suggest that SHIP OA
emissions were more weakly absorbing than moderately absorbing.

TRAF emissions were treated as very weakly absorbing in Cases 1, 2 and 3 and as weakly absorbing in Case 4. Interestingly, the highest TRAF *k* were obtained for URB sites whereas much lower *k* were obtained at REG and SUB sites. At URB sites, Cases 1, 2 and 3 provided *k* values in the upper range of the very weakly category (0.04; cf. Table 3) with very low std suggesting that TRAF emissions were weakly absorbing rather than very weakly absorbing. In fact, Case 4 provided a TRAF
*k* value of 0.06 that lied in-between the lower and upper limit of the category weakly absorbing. At both SUB and REG sites, the obtained TRAF *k* values were much lower compared to URB sites. This result is consistent with recent evidences that OA from traffic at urban sites can be an important source of brown OA (e.g., Ho et al., 2023). Moreover, traffic emissions are not expected to be primarily local at REG and SUB sites, thus likely contributing to the observed reduced TRAF *k* in these two settings due to physico-chemical OA processes as dilution and photobleaching.

OTHR emissions were treated as very weakly absorbing for all the considered cases (1-4). The optimization process provided very low *k* values confirming the very low absorption properties of OA particles emitted by OTHR sources. Note that higher OTHR *k* values were obtained at URB sites compared to REG and SUB sites where the obtained *k* were very close to the lower *k* value in the category very weakly (cf. Table 3).

Finally, and consistently with what was commented above, for the ALL case the obtained *k* followed the following order:
GFAS>RESI>TRAF>SHIP>OTHR.





**Table 5.** Statistical summary of mean and standard deviation (std) values for the optimization of the imaginary refractive index across the three settings (REG, SUB, URB) and the four cases (C1, C2, C3, C4). Each case corresponds to different assumptions detailed in Table 3, analyzed across five OA sources: GFAS, RESI, SHIP, TRAF, and OTHR. The row labeled "ALL" represents the averaged values across all stations for scenario C4.

| | | GFAS | | RESI | | SHIP | | TRAF | | OTHR | |
|---|---|---|---|---|---|---|---|---|---|---|---|
| | Metric | mean | std | mean | std | mean | std | mean | std | mean | std |
| Setting | Case | | | | | | | | | | |
| REG | Case 1 | 0.0619 | 0.0396 | 0.0458 | 0.0469 | 0.0196 | 0.0168 | 0.0093 | 0.0141 | 0.0015 | 0.0009 |
| | Case 2 | 0.0506 | 0.0434 | 0.0482 | 0.0448 | 0.0210 | 0.0071 | 0.0070 | 0.0139 | 0.0018 | 0.0010 |
| | Case 3 | 0.1241 | 0.0053 | 0.0476 | 0.0447 | 0.0527 | 0.0536 | 0.0068 | 0.0140 | 0.0014 | 0.0005 |
| | Case 4 | 0.0615 | 0.0387 | 0.0451 | 0.0462 | 0.0182 | 0.0150 | 0.0122 | 0.0132 | 0.0014 | 0.0008 |
| SUB | Case 1 | 0.0409 | 0.0154 | 0.0409 | 0.0154 | 0.0049 | 0.0000 | 0.0011 | 0.0000 | 0.0014 | 0.0005 |
| | Case 2 | 0.0403 | 0.0148 | 0.0403 | 0.0148 | 0.0191 | 0.0017 | 0.0011 | 0.0000 | 0.0013 | 0.0003 |
| | Case 3 | 0.1219 | 0.0000 | 0.0395 | 0.0135 | 0.0181 | 0.0000 | 0.0011 | 0.0000 | 0.0011 | 0.0000 |
| | Case 4 | 0.0398 | 0.0146 | 0.0398 | 0.0146 | 0.0049 | 0.0000 | 0.0049 | 0.0000 | 0.0012 | 0.0001 |
| URB | Case 1 | 0.0924 | 0.0354 | 0.0635 | 0.0535 | 0.0418 | 0.0481 | 0.0354 | 0.0000 | 0.0098 | 0.0109 |
| | Case 2 | 0.0998 | 0.0238 | 0.0635 | 0.0534 | 0.0445 | 0.0456 | 0.0354 | 0.0000 | 0.0092 | 0.0102 |
| | Case 3 | 0.1229 | 0.0017 | 0.0634 | 0.0535 | 0.0527 | 0.0599 | 0.0345 | 0.0015 | 0.0091 | 0.0110 |
| | Case 4 | 0.0934 | 0.0291 | 0.0625 | 0.0551 | 0.0314 | 0.0458 | 0.0579 | 0.0078 | 0.0081 | 0.0105 |
| ALL | Case 4 | 0.0640 | 0.0342 | 0.0481 | 0.0388 | 0.0182 | 0.0231 | 0.0218 | 0.0229 | 0.0030 | 0.0052 |

Based on the aforementioned $k$ optimization process, Case 4 was considered as the most appropriate to simulate absorption at the twelve measurement sites.

Figure 5 shows the optimized $k$ at 370 nm for total and individual components of the OA at the different monitoring sites. Two different approaches were used to obtain the results. We determined the optimal $k$ either by individual station (stn) or by combining all data points from every site (all). The figure illustrates the results for both Case 4 (based on source contribution) and Case 5 (a single value for all).

The optimization based on the total OA mass per station, Case 5 (by stn), results in $k$ values ranging from 0.005 (MSY site) to 0.07 (PAY site), which are representative of a very weakly to weakly/moderately absorbing OA (Table 3). In fact, 0.07 also falls within the lower range of the moderately category. A result of the overlap of $k$ ranges at 370 nm as mentioned before.

For REG stations, the optimized $k$ are notably low reflecting the importance of the contribution of biogenic SOA at these sites. MSY stands out with the lowest $k$ value of 0.005, which can be explained by the large contribution to OA from biogenic SOA (over 50%) characterized by very low absorption properties (In't Veld et al., 2023; Nakayama et al., 2010, 2012). Similarly, the low $k$ value (0.01) derived for HYY also indicates a high presence of biogenic SOA (Chen et al., 2022). Among regional



stations, the highest OA $k$ were obtained for IPR (0.03) and PAY (0.03) likely reflecting the influence of agricultural and
domestic biomass burning sources at these sites and the strong accumulation of absorbing POA in winter (see Figure 4) (Lanz
et al., 2010; Bozzetti et al., 2016; Daellenbach et al., 2017; Wolf et al., 2017).

In suburban environments, represented by KRA, SIR, and DEM, the $k$ values suggested a mix of urban influence and regional
aerosol contributions. For KRA, a pollution hotspot, a $k$ value of 0.02 was derived. A moderately absorbing environment likely
due to coal combustion, shipping activities from river cruise boats, and household heating emissions (Casotto et al., 2023;
Skiba et al., 2024). On the other hand, SIR and DEM present $k$ values of 0.01 and 0.02, respectively, are representative of
environments that mix local urban emissions with regional air masses.

For urban stations such as HEL, MAR and BCN, weakly to moderately absorbing OA are derived, reflecting the diverse
nature of urban emissions. Both HEL and BCN have similar $k$ (0.0284 and 0.0262), while MAR presents a notably higher
value (0.0494). The latter attributed to the harbor activities and local industrial emissions at that site (e.g., Chazeau et al.,
530   2022).

Comparing the results with the single $k$ derived from Case 5 (all) (dark green bars in Fig. 5), which corresponds to $k$ of 0.02,
suggests that OA particles across the different environments can be described on average as weakly absorbing particles. This
result is attributed to the averaging effect of combining highly absorbing components with those that are very weakly absorbing
into a single, undifferentiated category. Consequently, the optimization in this case likely masks the variability in absorption
strengths of individual OA components observed across Europe and may introduce biases in model absorption estimates.

Now, we analyze the results using the granular data provided by each individual station and emission source, Case 4 (by
stn), wherein the optimization is performed independently at each site (blue bars in Fig. 5) yielding different $k$ values for each
source. The observed variability for each component is as follow: GFAS ranges from 0.03 to 0.13, RESI from 0.008 to 0.13,
SHIP from 0.005 to 0.08, TRAF from 0.005 to 0.07, and OTHR from 0.001 to 0.02. These variations are associated with
the different environmental conditions of each station, within the limitations of our methodology to properly describe each
environment and the fact that sources that are negligible at a specific site introduces additional complexity in deriving a robust
estimate of $k$ for them.

Among regional background stations, PAY stands out showing the highest $k$ for GFAS (0.13) and RESI (0.13) components,
suggesting the significant influence of sources such as residential heating, which was previously observed by Ciarelli et al.
(2016), or the strong impact of regional events such as wildfires or agricultural waste burning. In contrast, RIG shows the
lowest $k$ for GFAS (0.03), while MSY presents the lowest value for RESI (0.008). As our optimization of $k$ relies on comparing
absorption measurements with modeled OA source components (assuming ranges of $k$ from different sources), there exists a
inherent relationship between mass contribution and absorption. This implies that although $k$ is independent of mass, sector
contribution could assign higher/lower $k$ values depending on the model uncertainties. Therefore, at regional sites like MSY,
the lower $k$ values for RESI may indicate reduced residential activity as shown by Pandolfi et al. (2014) and Navarro-Barboza
et al. (2024).



Suburban stations such as SIR, KRA, and DEM show consistent results across components, with $k$ values for sources like GFAS and RESI significantly higher than for SHIP, TRAF, and OTHR. The $k$ values for RESI and GFAS in DEM are nearly twice as high as those derived in SIR and KRA, indicating a mix of urban influence and regional aerosol contributions.

Conversely, urban stations, including HEL, MAR, and BCN, present significant variability, reflecting the diverse nature of urban emissions. MAR shows the highest $k$ for GFAS and RESI (0.13), suggesting a strong contribution to absorption from these sources. Notably, TRAF (0.06) is more significant in these urban stations compared to regional and suburban locations. In HEL, $k$ from shipping indicates a non-negligible contribution from this source, while in BCN, OTHR emerges as more significant (0.02), highlighting the complex and varied nature of urban aerosol sources.

Finally, Case 4 (all) represented by orange bars in Figure 5 involved the optimization by aggregating data from all stations while still differentiating between sources. The resulting $k$ values were 0.06 for GFAS, 0.04 for RESI, 0.06 for SHIP, 0.005 for TRAF, and 0.0011 for OTHR. These results are consistent with those reported in the literature for instance, Feng et al. (2013) derived a $k$ of 0.075 for moderately absorbing BrC from biomass burning at 350 nm, which is in agreement with our optimization results for GFAS. Similarly, the $k$ value for SHIP is consistent with Corbin et al. (2018), who reported a comparable value of 0.045 at 370 nm for this source. The optimized value for RESI is also consistent with literature, considering its association with biofuel combustion. Our estimation for TRAF, however, appears to be more than two times higher than the value found in the literature for this source (0.002 at 365 nm for octane combustion as reported by Hossen et al. (2023)). Nonetheless, we have observed considerable variability among URB sites, highlighting the intricate nature of urban environments. OTHR is identified as the least absorbing source, which is a reasonable outcome given its categorization.





**Figure 5.** Comparative optimization of $k$ at 370 nm for various OA sources, illustrating the results of two optimization strategies: Case 4, which optimize $k$ values for five distinct OA sources including, fires (GFAS), TRAF, SHIP, RESI, and OTHR, and Case 5, which considers the total OA and optimizes $k$ by aggregating data across all stations. Both cases present two approaches - one with $k$ values optimized for each station and another with a single k value derived from all stations combined.

In general, our findings agree well with values reported in the literature for specific sources. For instance, Pani et al. (2021) identified a $k$ of 0.12 at 370 nm for biomass burning closely aligned with the site most influenced by this source such as PAY. Furthermore, the SHIP component in our study shows a significant $k$, approaching to Corbin et al. (2018) who derived a value of 0.045 at 370 nm. Conversely, our TRAF results show a variation of $k$ from 0.005 to 0.07, falling in the upper range reported





in the literature for specific sources such as 0.002 at 365 nm for BrC from octane combustion (Hossen et al., 2023), or 0.027

for propane, 0.006 for diesel, 0.00074 for gasoline combustion reported at 550 nm (Lu et al., 2015).

### 3.3 OA Absorption results

In this section, we build upon the *k* values obtained in the previous section to analyze the OA absorption derived from model simulation and its annual variability. We explore the impact of using results obtained from aggregating all data (all) or exploiting the full granularity of the model and the observational data set (by stn).

#### 3.3.1 Absorption with optimized *k* by aggregating all data

Figure 6 shows a scatter plot of the observed versus modeled OA absorption at each monitoring site. The plot uses results from Case 4 (all) and Case 5 (all), where *k* values were obtained by using all the monitoring stations combined. To calculate total absorption, Case 4 uses individual *k* values tailored to specific OA components: 0.0571 for fires, 0.0403 for RESI, 0.0571 for SHIP, 0.0049 for TRAF, and 0.0011 for OTHR. Meanwhile, Case 5 applies a single *k* value of 0.0187 across all sources.

The key difference between the cases is that Case 4 includes a detailed source apportionment, discerning five different OA sources, while Case 5 considers the total OA without source differentiation. Case 5 represents the common approximation adopted by models in the literature (Takemura et al., 2005; Donner et al., 2011; Tegen et al., 2019) to describe the optical properties of aerosol components, where a single refractive index is assigned to each aerosol component. Conversely, Case 4 introduces the refinement at the source level to investigate the benefit of exploiting the source contribution to describe

absorption. Additionally, Figure 7 shows the time series of the absorption of OA at 370 nm simulated in Case 4 (all) and Case 5 (all) and the observational data for each monitoring site.

In Case 4 (dark green circles in Fig. 6), the correlation coefficients (r) range from 0.34 to 0.74 across all stations and the fractional bias (FB) values vary widely from -70% to 107%. Regional stations (OPE, PAY, and IPR), generally exhibit high correlations (>0.6) and a tendency to overestimate. Specifically, in MSY, the overestimation persists consistently throughout the

year, as shown in Figure 7 orange. Conversely, in HYY, this overestimation is predominantly observed during the initial months, while in RIG, it becomes noticeable towards the end of the year. This pattern might not necessarily reflect higher absorption but could be due to overrepresentation of absorbing sources in these sites, that are characterized by dominant contribution of biogenic SOA typically with little or no absorption. This tendency could be attributed to the considerable influence of secondary aerosol contributions. For instance, SHIP *k* value could be remarkably high, making it comparable to the values attributed to

fires and potentially biasing the overall absorption metric at some of these sites.

Suburban stations such as SIR, KRA, and DEM show good r values, specifically 0.74, 0.57, and 0.67, respectively. SIR stands out with a small FB of 3.8%, reflecting a very accurate representation of observed absorption. However, the model tends to underestimate the OA absorption for KRA and DEM, a phenomenon that is particularly pronounced at KRA, where absorption values exceed 50 $Mm^{-1}$ at 370 nm. This underestimation is likely linked to the increased emissions from households

and the energy industry at KRA site, as reported by Zgłobicki and Baran-Zgłobicka (2024), and the limitations previously highlighted in Section 3.1. The underestimation is predominantly observed during winter for KRA and DEM, whereas for SIR,



the underestimation occurs mainly during warm months, as illustrated in Figure 7. These seasonal discrepancies indicate that while the emission inventory (CAMS-REG-APv4.2) effectively captures the emissions near the SIR station, it may not fully account for the increased emissions during winter at KRA and DEM.

The urban stations HEL, MAR, and BCN show very different model performances. On the one hand, MAR is characterized by a substantial underestimation with low correlation value (0.49) and a notably negative FB of -82.3%. Conversely, BCN exhibits significant overestimation in the first months of the year, likely related to attribution issues with residential, traffic, and shipping emissions (Navarro-Barboza et al., 2024) as discussed in Section 3.1. On the other hand, HEL shows good agreement with observations (r = 0.51, FB = -7.5%).

In Case 5, where no source-specific components for OA are considered, correlation generally degrades compare to Case 4, with r values ranging from 0.28 to 0.65 across stations. FB values show again a wide variability, extending from a strong underestimation at a suburban site such as KRA (FB = -113%), particularly observed during winter, to an overestimation at a regional site like OPE (FB = 42%), mainly observed during summer and in MSY throughout the year.

     The differences between Case 4 and Case 5 highlight the benefit of using source specific $k$ in simulating OA absorption.
While Case 5 represents the common practice in atmospheric modeling of using a unique refractive index to describe aerosol optical properties, Case 4 provides an additional refinement that even improves seasonality, as illustrated in Figure 7 where Case 4 (all) follows increased absorption during winter and lower values during warmer months, while Case 5 (all) introduces important biases. Notably, OPE, SIR or HYY show substantial overestimation during summertime in Case 5 due to imposing an overly absorbing biogenic SOA, an issue that is effectively resolved in Case 4.





**Figure 6.** Daily mean observed versus modeled OA absorption for Case 4 (all) and Case 5 (all) scenarios, with the imaginary refractive index (*k*) values derived from aggregated data across all monitoring stations. Each panel includes the correlation coefficients (r) and fractional bias (FB).





**Figure 7.** Time series of the OA absorption at 370 nm across the 12 monitoring stations. Lines plot observed absorption (Obs) in red and modeled absorption for Case 4 (all) (orange line) and Case 5 (all) (dark green line). *k* values derived by combining data from all stations. Each set of panels corresponds to one season of the year (DJF: December-January-February, MAM: March-April-May), JJA: June-July-August), and SON: September-October-November).

In this regard, Figure 8 shows the mean seasonal variations in OA absorption coefficient at 370 nm simulated by MONARCH for the surface level in Europe. Both the total absorption coefficient and the source contributions are shown. The results are calculated based on optimized *k* for each OA source derived from Case 4 (all) aggregating data from all stations.

The total absorption shows the combined effect from all sources, with higher absorption observed in Central and Eastern Europe. Particularly relevant are the hot spots in the Po Valley (Italy), some regions in central and southern Poland and Roma-

nia reaching absorption coefficient values above 20 $Mm^{-1}$. RESI stands as the dominant contributor to total OA absorption, especially during the colder seasons (DJF, SON). In winter, the mean total absorption in Europe is around 0.9 $Mm^{-1}$, with residential sources explaining 80% of this (0.7 $Mm^{-1}$). Absorption shows a clear seasonality across Europe as seen in Figure 7. During summer, RESI contribution to light absorption at 370 nm in Europe reaches its minimum value, with a mean absorption coefficient of 0.1 $Mm^{-1}$, representing 28% of the summer OA absorption in Europe. Spring and Autumn periods





are characterized by still significant levels of absorption of 0.6 $Mm^{-1}$ and 0.7 $Mm^{-1}$ on average, respectively, dominated by RESI.

The second major source of absorption in our simulations is attributed to GFAS, particularly important during events in summer (0.15 $Mm^{-1}$ on average and maximum mean values above 10 $Mm^{-1}$). The mean absorption over Europe from fires is lower compared to residential sources but surface dominates in certain areas, especially in northern and southeast Europe, 640 where strong fires were detected by the satellite product. Notably, GFAS increases the background absorption in Europe with absorption values around 0.1 to 1 $Mm^{-1}$. Spring and Autumn are important wild fire seasons in eastern Europe, where large regions are affected by notably high GFAS absorption.

Another major contributor to absorption appears to be SHIP, with high absorption levels reaching values close to 5 $Mm^{-1}$ identified along major shipping routes, such as the English Channel, North Sea, and Mediterranean Sea. In this sense, coastal 645 regions are affected by this source, and the Mediterranean seas exhibit significant increases in absorption compared to inland areas. Some contributions also visible along river routes in eastern Europe. Shipping sources contribute relatively little to mean absorption in Europe, with the highest values in spring (0.02 $Mm^{-1}$).

Traffic-related absorption coefficients at 370 nm are relatively low compared to residential and shipping sources. There is a slight increase in absorption in urban areas and major transportation corridors, with minor seasonal variability in this source. 650 The traffic sources show their highest mean absorption coefficient in the surface layer in winter (0.04 $Mm^{-1}$), dominant in central Europe. Absorption from other sources is generally low across Europe, with minor seasonal variations. The highest mean absorption from OTHR occurs in autumn (0.08 $Mm^{-1}$).

### 3.3.2 Absorption with optimized *k* by station

Ultimately, a closer alignment of modeled absorption coefficients with measurements can be achieved by optimizing *k* for 655 each individual monitoring site. Determining a specific *k* for each site and emission source gives detailed information on the absorption characteristics of the site environments. Nevertheless, this detailed information is too location-specific to be utilized in atmospheric models. Regardless, the analysis offers valuable insights into the strengths and limitations of *k* discussed in the previous section for modeling applications.

Figure 9 shows scatter plots of modeled vs observed data. Moreover, Figure 10 depicts the time series of absorption at 370 660 nm for the twelve monitoring stations for these new two cases, Case 4 (blue line) and Case 5 (light green line).

Looking at the results for Case 4 (by stn) (blue dots in Figure 9), and comparing them with those of Case 4 (all) in Figure 6 (orange dots), we observe similar r values ranging from 0.36 to 0.74 in both figures. However, optimizing per station significantly improves the FB values at all sites (see Table S2). The use of station-specific and category-specific *k* values appears to substantially reduce bias, as expected. Among regional background stations, MSY stands out improving results from a strong 665 overestimation (FB: 107%, assuming a constant *k* for each OA source over all aggregated data, orange line in Figure 7) to a slight underestimation, as shown in Figure 10 blue line. Both winter and summer periods are adjusted in this station, a good example of the added value of refining the characterization of sources for a specific environment. Other stations such as PAY, RIG and IPR also show consistent improvements compared with Case 4 (all).





**Figure 8.** Seasonal source contribution to OA surface absorption at 370nm in Europe for the year 2018. MONARCH simulation using the optimized *k* for each OA source derived from Case 4 (all) shown in orange bars in Figure 5. The seasonal breakdown is as follows: the first column represents December-January-February (DJF), the second column represents March-April-May (MAM), the third column represents June-July-August (JJA), and the fourth column represents September-October-November (SON). The rows indicate different sources, with the top row showing the total, followed by residential, shipping, fires, traffic, and other sources in descending order.





At suburban sites, the model reasonably captures the absorption at DEM, particularly noting the well-captured peaks during
winter. Interestingly, KRA persistently shows large underestimations during colder months (FB = -103%). This clearly indicates
the limitations in the way emissions are represented in the inventory used in this study.

Urban stations show different responses to the refined optimization. Modeled absorption improves compared to Case 4 (all)
in BCN and MAR, while in HEL a slight degradation during the colder months occurs. In general, the model captures the peaks
throughout the year. The fire event identified by the model in July in MAR is simulated with a significant overestimation. This
could indicate a limitation in the fire emissions data for this specific event.

In Case 5 (by stn), a unique station-specific $k$ value is employed without source differentiation. When comparing r values
with the approach that utilizes a single $k$ across all stations (as depicted in Figure 6), consistent results are found with a clear
improvement in the FB across all stations as also shown in Case 4. This tailored approach generally leads to underestimations
of the absorption coefficient at 370 nm in the surface layer at most of the twelve stations studied. In contrast, absorption is
notably overestimated at PAY (regional background station) from April to October. The limited number of observations used
in the optimization step at this site, mainly dominated by few winter measurements, could explain the overestimated $k$ value.

Overall, the insights gained from this analysis recommends adopting approaches towards refining $k$ values used in models to
better represent the unique characteristics of each station and emission source. The tailored approach of Case 4, with its more
granular differentiation of $k$ values, contributes in the characterization of the different sites investigated in this work.





**Figure 9.** Similar to Figure 6, but with *k* derived for each station.







**Figure 10.** Similar to Figure 7 but the *k* optimization process was done at each station.

## 4 Summary and conclusions

In this study, we explored the refractive imaginary index (*k*) for OA components from different sources, including fires (GFAS), residential (RESI), shipping (SHIP), traffic (TRAF), and others emissions (OTHR). Our analysis relied on 12 distinct monitoring stations throughout Europe, representative of regional, suburban, and background urban environments to capture the diverse environmental conditions prevalent at each site. Using a synergistic approach that combined both modeling and observational techniques, we conducted year-long simulations covering the year 2018. These simulations were designed to evaluate the OA mass concentrations using the MONARCH chemical transport model and constrain the OA light absorbing properties.

We used OA mass concentrations derived from Aerosol Chemical Speciation Monitor (ACSM) and filter based measurements, and BrC absorption coefficients data retrieved from aethalometer measurement at 370 nm as reference datasets.

Furthermore, we developed an offline optical tool designed to estimate the OA absorption coefficient from the OA mass concentrations calculated from the model. We used the optical tool to derive *k* values at 370 nm that minimize the error





with the absorption measured at the 12 monitoring stations in Europe using a Sequential Least Squares algorithm (SLSQP). Bounding values based on Saleh (2020) were imposed to derive the optimized $k$ for each station and emission source. The analysis combined optimizations using all the observational data aggregated, exploring the granular information available (source contribution estimates at each monitoring site) and intermediate combinations.

Overall the MONARCH model shows good performance in simulating OA mass concentrations, with good agreement between measured and modeled concentrations for the majority of the stations. The statistical evaluation indicates that most of the stations met the evaluation benchmark defined by Emery et al. (2017) throughout the year. The model best performance is identified during spring. Notably, residential emissions (accounting for domestic heating, cooking and water heating) emerge as a predominant source of OA mass concentrations during colder months. This share may be biased in some European regions

by an overrepresentation of residential heating emissions in the CAMS-REG-AP_v42 inventory, as was already highlighted by Navarro-Barboza et al. (2024). The second most relevant contribution to OA mass concentrations comes from secondary organic aerosols (SOA) in most of the stations, particularly during warmer periods at regional sites such as MSY, HYY, OPE, RIG, and PAY. SOA is slightly underestimated in summer, probably due to low SOA yields used in the simulation for biogenic sources. Furthermore, shipping emissions (SHIP) play a significant role at near port stations. Despite the limitations identified,

the model effectively captures specific events like wild fires and high pollution episodes, demonstrating its ability to reproduce episodic events. Traffic-related emissions contributes to the lesser exent to OA mass concentrations.

The optimization of the imaginary part of OA refractive index $k$ at the monitoring stations has underscored the complex and dynamic nature of OA optical properties, which are influenced by emission sources and environmental conditions. We derived $k$ values at 370 nm for total OA that ranged from 0.005 (weakly absorbing) to 0.068 (weakly to moderately absorbing),

highlighting the significant variability in OA absorption properties across Europe. This observation aligns with existing knowledge that OA properties are not static, but vary depending on composition, source, and atmospheric age (Cappa et al., 2011). Regional background stations such as MSY and HYY exhibited the lowest $k$ values, consistent with the predominant biogenic SOA in these regions.

The approach we used to optimize $k$ for various OA sources, such as fires (GFAS), residential (RESI), shipping (SHIP),

traffic (TRAF), and others (OTHR), highlights the added value of source apportionment in precisely characterizing OA optical properties. Our results revealed the significant impact of local and regional emissions on $k$ values. For example, at the PAY station, which is characterized by a regional background environment, elevated $k$ values were derived for wild fires and residential combustion sources (e.g., biomass, coal), which aligns with the known strong absorption characteristics of biomass burning aerosols (Zhang et al., 2020b; Pani et al., 2021). This is further supported by the evidence suggesting that BrC from

biomass burning can undergo significant dark chemical processing, affecting over 70% of OA from this source (Kodros et al., 2020). Conversely, other regional stations such as HYY, OPE, RIG, and MSY were characterized by a dominance of SOA from biogenic emissions, reinforcing the role of biogenic sources in the low $k$ values derived (Zhao et al., 2015). In urban environments, traffic emissions emerged as a significant contributor to light absorption at 370 nm with a $k$ value (0.06) two times more than that reported for BrC from specific traffic-related emissions (e.g., propane, diesel, gasoline) (Hossen et al.,

2023; Lu et al., 2015), indicative of the complexity of urban environments. Additionally, stations near ports were found to have





a relevant SHIP contribution to OA optical properties, consistent with previous studies (Kapoor et al., 2023). In summary, we have derived specific ranges of $k$ values at 370 nm for various emission sources using all the granular information available in our study as follows:

- Biomass burning (GFAS): $k$ values range from 0.03 to 0.13. This broad range reflects the variability between stations close to and far from fire sources, highlighting the diverse impact of biomass burning on OA absorption and significance of bleaching processes.

- Residential sources (RESI): $k$ values range from 0.008 to 0.13. Highlighting the variability in residential activities and practices across different regions (e.g., extensive use of coal combustion in KRA).

- Shipping sources (SHIP): Near port areas, $k$ values range from 0.005 to 0.08. In some sites, the second most absorbing source identified in our study in terms of $k$ values.

- Traffic (TRAF): $k$ values range from 0.005 to 0.07 i.e., larger than found for specific traffic emission sources in the literature. This result indicate possible unaccounted processes contributing to enhance absorption in urban environments.

- other sources (OTHR): $k$ values range from 0.001 to 0.02. These sources include emissions from power generation, industry,solvents, aviation, waste treatment and disposal, agriculture, etc. and exhibit lower absorption properties indicating the varied influence of emissions on OA absorption.

The implementation of these source-specific $k$ values significantly enhances the agreement between modeled and observational data, improving the model performance compared to the use of a constant $k$ value to characterize OA absorption. This is actually a common practice in several atmospheric models adopting for instance $k$ value at 550 nm from 0.005 to 0.006 (Matsui and Mahowald, 2017; Tegen et al., 2019; Bozzo et al., 2020; Wang et al., 2013; Burgos et al., 2020).

This widespread modeling practice underscores the relevance of our findings, proposing a refined method for determining $k$ values that could improve the accuracy of future estimates of BrC radiative forcing.

In this sense, we computed and presented the source contribution to OA light absorption at the surface level in Europe based on the optimized $k$ derived from our analysis. Total absorption is the highest in Central and Eastern Europe, with notable hotspots in the Po Valley, Poland, and Romania. Residential sources are the dominant contributors to OA absorption, especially during the colder seasons (DJF, SON). In winter, RESI accounts for 80% of total absorption, a value that decreases to 28% in summer. Fires (GFAS) are the second major source, particularly during summer, while shipping (SHIP) contributes to high absorption levels along major routes, impacting coastal regions. Traffic-related absorption is relatively low, with minor seasonal variability, and presents its major contribution during winter in central Europe. Lastly, other sources (OTHR) have the lowest contribution to total absorption levels in Europe with reduced seasonal variation.

Our study also recognizes the limitations of current models and emission inventories, which may contribute to discrepancies between observed and modeled data. For instance, the underestimation of light absorption by OA could result from underpredicted biomass burning or biofuel emissions, as well as uncertainties in particle size and mixing state (Huang et al.,



2013). Additionally, the role of BrC may be underestimated in the study, as fossil fuel OA could also be light-absorbing (Lee et al., 2014). Future enhancements should aim at refining emission inventories and improving the representation of BrC and

SOA in atmospheric models. Moreover, this study underscores the need to increase observational measurements of BrC light absorption across Europe, which could help further constrain the effect of various sources on OA properties.

By establishing a link between laboratory measurements, field observations and modeling experiments, our study offers insights that could improve the representation of OA optical properties in global atmospheric models, thereby advancing our understanding of aerosol-climate interactions and their broader environmental implications.

*Author contributions.* HN, MP and OJ contributed to the conceptualization, design, and the analysis of the work. HN contributed to the emissions processing, conducting MONARCH runs and data processing. VO and HN contributed to the optical properties calculation. JR, MP, MV, AA, XQ, NP, MS, GC, JY, MI, MR, KE, SV, OZ, MG, BC, NM, AP, KD, ME, KL, TP, AT, JN, MA, HT, JNi, OF, JP, JPh, CH, NPa, AC, SC contributed to the provision of measurement data. APo contributed to data analysis. OJ, MP, XQ, AA contributed to acquiring funding. HN, MP and OJ prepared the manuscript with contributions from all co-authors.

*Competing interests.* At least one of the (co-)authors is a member of the editorial board of Atmospheric Chemistry and Physics.

*Acknowledgements.* Hector Navarro-Barboza was funded by grant PRE2018-084988 from the FPI programme by the Spanish Ministry of the Economy and Competitiveness. The research leading to these results has received funding from the Ministerio de Ciencia, Innovación y Universidades as part of the the BROWNING project (grant no. RTI2018-099894-B-I00 funded by MCIN/AEI/10.13039/501100011033 and by "ERDF A way of making Europe") and the Ministerio de Asuntos Económicos y Transformación Digital, Gobierno de España

as part of the CAIAC project (grant no. PID2019-108990PB-100). The research leading to these results has also received funding from the EU HORIZON-EUROPE under grant agreement n° 101056783 (FOCI project), the EU HORIZON2020 under grant agreement n° 821205 (FORCES project) and n° 101036245 (RI-URBANS project), and the Department of Research and Universities of the Government of Catalonia via the Research Group Atmospheric Composition (code 2021 SGR 01550 and 2021 SGR 00447). IDAEA-CSIC is a Centre of Excellence Severo Ochoa (Spanish Ministry of Science and Innovation, grant no. CEX2018-000794-S). FMI gratefully acknowledges

funding Research Council of Finland via the project Black and Brown Carbon in the Atmosphere and the Cryosphere (BBrCAC) (decision nr. 341271).

BSC researchers thankfully acknowledge the computer resources at Marenostrum and the technical support provided by Barcelona Supercomputing Center (RES-AECT-2022-1-0008, RES-AECT-2022-2-0003).



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
