# Peer review of "Characterization of Brown Carbon absorption in different European environments through source contribution analysis"

_EGUsphere, 2024_

## Community Comment (CC2)

As claimed by authors in the manuscript that "However, theoretical simulations have shown that the AAEBC can reasonably vary between 0.9 and 1.1 depending on the size and internal mixing of BC particles"

Lack and Langridge (2013) reviewed a range of field measurements of "encapsulated" BC and suggested an $AAE_{BC}$ of 1.1±0.3 (0.8-1.4), Luo et al. (2022) simulated variations of $AAE_{BC}$ and discussed key factors that influencing variations in $AAE_{BC}$ and found that the largest factor that influence the variations in $AAE_{BC}$ is the black carbon mass size distributions, and the $AAE_{BC}$ could even out the range of 0.8-1.4 (Figure 1a), larger than the range stated by the authors "However, theoretical simulations have shown that the $AAE_{BC}$ can reasonably vary between 0.9 and 1.1 depending on the size and internal mixing of BC particles (Bond et al., 2013; Lu et al., 2015, e.g.,)" . This could be verified by results from field measurements, for example, the probability distribution of $AAE_{880-950}$ and $AAE_{660-880}$ show in Figure 2a of Luo et al. (2022).

let's assume three cases and ignore the spectral dependence of AAEBC in this part.

The first case is that $AAE_{BC}$ equal to 0.9, however, the significant contribution of BrC made the fitted AAE is 0.95; The second case, is that $AAE_{BC}$ equal to 1.05, however, the contribution of BrC is negligible, which made the fitted AAE is still 1.05. The third case is $AAE_{BC}$ equal to 1.1, however the contribution of BrC resulted in the fitted AAE is also 1.15. Based on the method used by authors, the AAE of 0.95 would be chosen to represent $AAE_{BC}$, however would bias from the true average $AAE_{BC}$ of 1.02. This example tells us the 1st percentile of fitted AAE depends on the covariations of $AAE_{BC}$ and BrC contributions, it does not help acquire the average $AAE_{BC}$. The most important factor that influencing variations in $AAE_{BC}$ - black carbon mass size distribution and BrC absorptions are controlled by complex processes that is quite difficult to disentangle. The derived $AAE_{BC}$ lie between 0.9 and 1.1 does not make this method valid.

Authors argued that "Zhang et al. (2020) have reported an uncertainty of approximately 11% in the estimation of the BrC contribution to total absorption at 370 nm when using different $AAE_{BC}$ values ranging from 0.9 and 1.1". That being the case, using $AAE_{BC}$ of 1 is just fine, there is no need to derive AAEBC using a method seems reasonable. I agree with authors that sometimes signals at 950 nm can be very low, however that at 660 nm should be fine, I suggest that authors present the probability distribution of $AAE_{660-880}$ to show possible variation ranges of $AAE_{BC}$ and directly using the average $AAE_{660-880}$ to represent average $AAE_{BC}$ of each site might be more reasonable, because BrC absorption at 660 nm is also very small. With respect to spectral dependence of $AAE_{BC}$, Wang et al. (2018) found that the spectral dependence of $AAE_{BC}$ should be considered, however, the proposed method assume that BrC absorption is negligible which is not the real case, as stated by authors. Therefore, Luo et al. (2022) proposed an improved AAE ratio method considering both variations and spectral dependence of black carbon AAE to differentiate brown carbon (BrC) absorptions from total aerosol absorptions. They use $AAE_{880-950}$ to account for the variations embedded in $AAE_{BC}$ and the ratio $R_{AAE}(\lambda) = AAE_{BC,\lambda-880}/AAE_{BC,950-880}$ to take spectral dependence of $AAE_{BC}$ into account, not using $AAE_{880-950}$ to account for $AAE_{BC}$ as stated in responses of authors.

Therefore, the formula of deriving BrC($\lambda$) is :

$$\sigma_{BrC}(\lambda) = \sigma_a(\lambda) - \sigma_{BC}(880\ nm) \times \left(\frac{880}{\lambda}\right)^{AAE_{BC,950-880} \times R_{AAE}(\lambda)}$$

Let's move back to the BrC(370) calculation formula presented by authors:

$$\sigma_{BrC}(\lambda) = \sigma_a(\lambda) - \sigma_{BC}(880\,nm) \times \left(\frac{880}{\lambda}\right)^{AAE_{BC}}$$

Authors using $\sigma_{BC}(880\,nm)$ to derived $\sigma_{BrC}(\lambda)$, based on the definitions of AAE, the $AAE_{BC,\ \lambda-880}$ should be the focus. If using constant AAE$_{BC}$ derived through fitting BC absorptions at multi-wavelengths, would result in different uncertainties at different $\lambda$ values. Therefore, if we want to accurately retrieve for example, $\sigma_{BrC}(370)$, then we should focus on representing $AAE_{BC,\ 370-880}$ accurately. However, as simulated by Luo et al. (2022), $AAE_{BC,\ 370-880}$ would be much smaller than $AAE_{BC,\ 660-880}$ or $AAE_{BC,\ 880-950}$ and the ratio between RAAE(370) depends mostly on black caron mass size distributions (Figure 1b). The used R$_{AAE}$(370) in Luo et al. (2022) for deriving $\sigma_{BrC}(370)$ is 0.79, if this ratio holds for sites of this manuscript. Then $AAE_{BC,\ 370-880}$ should be less than 0.8, which I believe would result in non-negligible underestimations of $\sigma_{BrC}(370)$ if authors use $AAE_{BC}$ of 1 or other values to derive $\sigma_{BrC}(370)$.

In summary, I agree with authors that "This is a reasonable uncertainty considering the overall uncertainty of the AAE method. Moreover, also the modeling part presented in this work is prone to uncertainties and any change of AAE$_{BC}$ can add uncertainties that however lie well within the overall uncertainty of the approach presented in this manuscript".

Now that authors mentioned variations in AAE$_{BC}$ and tried to derived a reasonable one, we should discuss comprehensively the best way of deriving $\sigma_{BrC}(\lambda)$ on the basis of limited multiwavelength aerosol absorption measurements and deliver clearly to readers.

In summary, I suggest that authors using the average AAE$_{660-880}$ to represent AAE$_{BC}$ variations of different sites, and account for the spectral dependence by simulating a ratio R$_{AAE}$(370)=$AAE_{BC,370-880}/AAE_{BC,660-880}$ using typical black carbon mass size distributions in Europe on the basis of Mie theory. If not, at least discuss the potential uncertainties associated with the spectral dependence of AAE$_{BC}$ to deliver comprehensive understanding of $\sigma_{BrC}(\lambda)$ derivations that including latest advancements. Moreover, I want to highlight that considering the spectral dependence of AAE$_{BC}$ is quite important if we want to investigate the spectral dependence of BrC absorptions.

I greatly appreciate this manuscript, which synthesizes measurements from over ten sites in Europe. Therefore, it deserves thorough scrutiny and attention.

**References**:

Lack, D. A., and Langridge, J. M.: On the attribution of black and brown carbon light absorption using the Ångström exponent, Atmos. Chem. Phys., 13, 10535-10543, 10.5194/acp-13-10535-2013, 2013.

Luo, B., Kuang, Y., Huang, S., Song, Q., Hu, W., Li, W., Peng, Y., Chen, D., Yue, D., Yuan, B., and Shao, M.: Parameterizations of size distribution and refractive index of biomass burning organic aerosol with black carbon content, Atmos. Chem. Phys., 22, 12401-12415, 10.5194/acp-22-12401-2022, 2022.

Wang, J., Nie, W., Cheng, Y., Shen, Y., Chi, X., Wang, J., Huang, X., Xie, Y., Sun, P., Xu, Z., Qi, X., Su, H., and Ding, A.: Light absorption of brown carbon in eastern China based on 3-year multi-wavelength aerosol optical property observations and an improved absorption Ångström exponent segregation method, Atmos. Chem. Phys., 18, 9061-9074, 10.5194/acp-18-9061-2018, 2018.

---

## Author Comment (AC2)

**Response to PhD Ye Kuang: *Characterization of Brown Carbon absorption in different European environments through source contribution analysis**

November 18, 2024

As claimed by authors in the manuscript that

> "However, theoretical simulations have shown that the $AAE_{BC}$ can reasonably vary between 0.9 and 1.1 depending on the size and internal mixing of BC particles."

Lack and Langridge (2013) reviewed a range of field measurements of "encapsulated" BC and suggested an $AAE_{BC}$ of $1.1 \pm 0.3$ (0.8–1.4). Luo et al. (2022) simulated variations of $AAE_{BC}$ and discussed key factors influencing variations in $AAE_{BC}$ and found that the largest factor that influences the variations in $AAE_{BC}$ is the black carbon mass size distribution, and the $AAE_{BC}$ could even be out of the range of 0.8–1.4 (Figure 1a), which is larger than the range stated by the authors:

> "However, theoretical simulations have shown that the $AAE_{BC}$ can reasonably vary between 0.9 and 1.1 depending on the size and internal mixing of BC particles" (Bond et al. (2013); Lu et al. (2015), e.g.).

This could be verified by results from field measurements. For example, the probability distribution of $AAE_{880-950}$ and $AAE_{660-880}$ is shown in Figure 2a of Luo et al. (2022).

Let's assume three cases and ignore the spectral dependence of $AAE_{BC}$ in this part. The first case is that $AAE_{BC}$ is equal to 0.9; however, the significant contribution of BrC made the fitted AAE 0.95. The second case is that $AAE_{BC}$ is equal to 1.05; however, the contribution of BrC is negligible, which made the fitted AAE still 1.05. The third case is that $AAE_{BC}$ is equal to 1.1; however, the contribution of BrC resulted in the fitted AAE of 1.15. Based on the method used by the authors, the AAE of 0.95 would be chosen to represent $AAE_{BC}$, but it would bias from the true average $AAE_{BC}$ of 1.02.

This example tells us the 1st percentile of fitted AAE depends on the covariations of $AAE_{BC}$ and BrC contributions; it does not help acquire the average $AAE_{BC}$. The most important factor influencing variations in $AAE_{BC}$—black carbon mass size distribution—and BrC absorptions are controlled by complex processes that are quite difficult to disentangle. The derived $AAE_{BC}$ lying between 0.9 and 1.1 does not make this method valid.

Authors argued that

> "Zhang et al. (2020) have reported an uncertainty of approximately 11% in the estimation of the BrC contribution to total absorption at 370 nm when using different $AAE_{BC}$ values ranging from 0.9 and 1.1."

That being the case, using $AAE_{BC}$ of 1 is just fine; there is no need to derive $AAE_{BC}$ using a method that seems reasonable. I agree with the authors that sometimes signals at 950 nm can be very low; however, those at 660 nm should be fine. I suggest that the authors present the probability distribution of $AAE_{660-880}$ to show possible variation ranges of $AAE_{BC}$ and directly

use the average $AAE_{660\text{-}880}$ to represent average $AAE_{BC}$ at each site, which might be more reasonable because BrC absorption at 660 nm is also very small.

With respect to the spectral dependence of $AAE_{BC}$, Wang et al. (2018) found that the spectral dependence of $AAE_{BC}$ should be considered. However, the proposed method assumes that BrC absorption is negligible, which is not the real case, as stated by the authors. Therefore, Luo et al. (2022) proposed an improved AAE ratio method considering both variations and the spectral dependence of black carbon AAE to differentiate brown carbon (BrC) absorptions from total aerosol absorptions. They use $AAE_{880\text{-}950}$ to account for the variations embedded in $AAE_{BC}$, and the ratio $R_{AAE}(\lambda) = \frac{AAE_{BC,\lambda-880}}{AAE_{BC,950-880}}$ to take the spectral dependence of $AAE_{BC}$ into account, not using $AAE_{880\text{-}950}$ to account for $AAE_{BC}$ as stated in the responses of the authors.

Therefore, the formula for deriving $\sigma_{\mathrm{BrC}}(\lambda)$ is:

$$\sigma_{\mathrm{BrC}}(\lambda) = \sigma_a(\lambda) - \sigma_{\mathrm{BC}}(880\,\mathrm{nm}) \times \left(\frac{880}{\lambda}\right)^{AAE_{BC,950-880} \times R_{AAE}(\lambda)}$$

Let's move back to the BrC(370) calculation formula presented by the authors:

$$\sigma_{\mathrm{BrC}}(\lambda) = \sigma_a(\lambda) - \sigma_{\mathrm{BC}}(880\,\mathrm{nm}) \times \left(\frac{880}{\lambda}\right)^{AAE_{BC}}$$

Authors used $\sigma_{\mathrm{BC}}(880\,\mathrm{nm})$ to derive $\sigma_{\mathrm{BrC}}(\lambda)$. Based on the definitions of AAE, the $AAE_{BC,\lambda-880}$ should be the focus. If using a constant $AAE_{BC}$ derived through fitting BC absorptions at multiple wavelengths, it would result in different uncertainties at different $\lambda$ values. Therefore, if we want to accurately retrieve, for example, $\sigma_{\mathrm{BrC}}(370)$, then we should focus on representing $AAE_{BC,370-880}$ accurately. However, as simulated by Luo et al. (2022), $AAE_{BC,370-880}$ would be much smaller than $AAE_{BC,660-880}$ or $AAE_B, 880-950$, and the ratio of $R_{AAE}(370)$ depends mostly on black carbon mass size distributions (Figure 1b). The used $R_{AAE}(370)$ in Luo et al. (2022) for deriving $\sigma_{\mathrm{BrC}}(370)$ is 0.79; if this ratio holds for the sites of this manuscript, then $AAE_{BC,370-880}$ should be less than 0.8, which I believe would result in non-negligible underestimations of $\sigma_{\mathrm{BrC}}(370)$ if authors use $AAE_{BC}$ of 1 or other values to derive $\sigma_{\mathrm{BrC}}(370)$.

In summary, I agree with the authors that

> "This is a reasonable uncertainty considering the overall uncertainty of the AAE method. Moreover, the modeling part presented in this work is prone to uncertainties, and any change of $AAE_{BC}$ can add uncertainties that, however, lie well within the overall uncertainty of the approach presented in this manuscript."

Now that the authors have mentioned variations in $AAE_{BC}$ and tried to derive a reasonable one, we should comprehensively discuss the best way of deriving $\sigma_{\mathrm{BrC}}(\lambda)$ on the basis of limited multi-wavelength aerosol absorption measurements and deliver this clearly to readers.

In summary, I suggest that the authors use the average $AAE_{660\text{-}880}$ to represent $AAE_{BC}$ variations at different sites and account for the spectral dependence by simulating a ratio $R_{AAE}(370) = \frac{AAE_{BC,370-880}}{AAE_{BC,660-880}}$ using typical black carbon mass size distributions in Europe on the basis of Mie theory. If not, at least discuss the potential uncertainties associated with the spectral dependence of $AAE_{BC}$ to deliver a comprehensive understanding of $\sigma_{\mathrm{BrC}}(\lambda)$ derivations that include the latest advancements. Moreover, I want to highlight that considering

**Response:** Here we considered a reasonable $AAE_{BC}$ range between 0.9 and 1.1 based on values obtained, for example, by constraining the $AAE_{BC}$ determination with $^{14}$C analysis ($AAE_{BC} = 0.9$ in Zotter et al. (2017), further confirmed by Blanco-Alegre et al. (2022) using aethalometer measurements in a road tunnel). It has also been stated in the literature that the AAE of externally mixed BC is approximately 1 for particles $< 50$ nm in diameter, and can range from 0.8 to 1.1 for diameters of 50–200 nm (Gyawali et al., 2009). Extreme AAE values for internally

mixed BC up to 1.7 have also been reported (Gyawali et al., 2009). Other studies have reported an average $AAE_{BC}$ of 1.1 (Lack and Langridge (2013) and references herein).

As stated in Lack and Langridge (2013), the extreme values ($0.55 < AAE_{BC} < 1.7$) that have been reported in the literature for very specific BC particles "are likely not common in the atmosphere for $BC_{Ext}$ and $BC_{Int}$, and serve here as extreme boundaries only."

Thus, the extreme $AAE_{BC}$ reported in the literature were associated with very specific BC particles that do not necessarily represent the heterogeneity of BC particles under ambient conditions. Some of these extreme $AAE_{BC}$ values were, for example, obtained from laboratory experiments or from theory.

As an example, we applied the range of $AAE_{BC}$ mentioned by Dr. Kuang ($0.8 < AAE_{BC} < 1.4$) to the Barcelona dataset. The Barcelona measurement station is highly affected by BC emissions from vehicles passing the busiest road of the city located 200 m from the measurement site. Previous studies conducted in Barcelona have shown a low contribution from biomass burning in the city (e.g., Via et al. (2021)). Via et al. (2021) reported an average contribution of biomass burning to OA mass concentration in Barcelona of 4% during 2017–2018. If an $AAE_{BC}$ of 0.8 is used, the BrC contribution to absorption at 370 nm reached 50% of total absorption (annual average), which is clearly too high for Barcelona. If an $AAE_{BC}$ of 1.4 is used, then a negative (-20%) BrC contribution to absorption is obtained. The first percentile of $R^2$-filtered AAE provided a value of 1 for Barcelona, which resulted in a very reasonable estimation of BrC contribution to absorption in Barcelona based on previous studies performed in the city (14%). In Ispra (located in the Po Valley in Northern Italy), an $AAE_{BC}$ of 0.8 would provide a BrC contribution to absorption close to 90% (too high), and an $AAE_{BC}$ of 1.4 would provide a BrC contribution to absorption of 10% (too low). Similarly, for Krakow (where the 1st percentile provided an $AAE_{BC}$ of 1.07), using 0.8 and 1.4 provided BrC contributions to absorption of 85% and 9%, respectively, which were, respectively, too high and too low. This small sensitivity study confirms that the extreme $AAE_{BC}$ values cannot be used to represent BC under ambient conditions and that $AAE_{BC}$ values closer to one provide the best estimation.

We would like to comment that the use of the $R^2$-filtered AAE frequency distribution (FD) for the determination of $AAE_{BC}$ has been published by Tobler et al. (2021) and Glojek et al. (2024). These authors estimated the $AAE_{BC}$ by visually inspecting the AAE frequency distributions and set the $AAE_{BC}$ somewhere in the very left tail of the FD. Here, we suggested a possible mathematical approach using the 1st percentile to avoid a too subjective determination of $AAE_{BC}$ from AAE data.

We also highlight that the first percentile represents conditions when the absorption is dominated by BC and not by both BC and OA. Consequently, the possible presence of very specific BC particles (causing too low or too high $AAE_{BC}$ in other studies) should be reflected in the experimental AAE values, and consequently in the 1st percentile. But this is not the case for the 12 sites used in the manuscript, where the 1st percentile provided $AAE_{BC}$ values mostly from 0.9 to 1.1.

We are aware that Luo et al. (2022) proposed an improved AAE ratio method using the AAE calculated from 880 and 950 nm to account for the spectral variation of $AAE_{BC}$. However, Dr. Kuang agrees with us that the absorption at 950 nm can be very noisy and that its systematic use at many sites (especially at regional/remote sites) cannot be guaranteed. Thus, a true harmonization of the attribution method at the 12 sites cannot be applied in this manuscript if the absorption at 950 nm is used. For this reason, Dr. Kuang suggests using the AAE calculated from 660 to 880 nm instead of from 880 to 950 nm. Thus, Dr. Kuang suggests applying the method proposed in Luo et al. (2022), but using a new wavelength pair (i.e., 660–880 nm). Consequently, there are two issues: one is that we would need to apply a methodology that has never been applied before in the literature using this new specific wavelength pair from aethalometer data. This is out of the scope of this manuscript. Second, the assumption that BrC particles do not absorb at 660 nm is not reasonable in many cases. Thus, exploring the

possible spectral dependence of $AAE_{BC}$, as done in Luo et al. (2022), or its modified version as suggested by Dr. Kuang, is not systematically possible at all the sites included in the manuscript. Moreover, the approach from Luo et al. (2022) implies the use of BC size distribution data that are not available at the measurement sites included in the manuscript. Dr. Kuang suggests using some typical BC size distribution for Europe, but this introduces an additional uncertainty that cannot be estimated considering that BC size distribution is highly variable. It should be noted that the aerosol particle size distribution was measured in Luo et al. (2022) and not assumed from other studies.

For the aforementioned reasons and in order to discuss the potential uncertainties associated with the attribution method, we modified the sentences from Line 152 as follows:

"... where $AAE_{BC}$ is the Absorption Ångström Exponent (AAE) of BC, which allows for the calculation of $b_{abs,BC(\lambda)}$ (in units of $Mm^{-1}$) from the measurements of $b_{abs,BC(\lambda)}$ at 880 nm assuming that BrC does not absorb at 880 nm (e.g., Qin et al. (2018)). The main source of uncertainty in equations 1 and 2 is the AAE assumed for BC. In many studies, a value of 1 was used (Liakakou et al. (2020); Tian et al. (2023); Cuesta-Mosquera et al. (2023), e.g.). However, theoretical simulations have shown that the $AAE_{BC}$ can reasonably vary between 0.9 and 1.1 depending on the size and internal mixing of BC particles (e.g., Bond et al. (2013); Lu et al. (2015)). Here we estimated the site-dependent $AAE_{BC}$ as the first percentile of the AAE frequency distribution. The AAE can be calculated from multi-wavelength (370, 470, 520, 590, 660, 880, and 950 nm) total absorption measurements as the linear fit in a log-log plot of the total absorption versus the measuring wavelengths. The effect of BrC absorption is to increase the AAE, and consequently, the first percentile of AAE represents conditions where the absorption is dominated by BC. In order to reduce the noise, the 1st percentile at each site was calculated from AAE values obtained from fits with $R^2 > 0.99$ (Tobler et al., 2021; Glojek et al., 2024). Other approaches used a combination of Mie theory and experimental data to explore the wavelength dependence of $AAE_{BC}$ and proposed an estimation of $b_{abs,BrC(\lambda)}$ based on the ratio between the AAE calculated from 370 to 520 nm and from 520 to 880 nm (Wang et al., 2018; Li et al., 2019). However, this methodology assumed that BrC particles do not absorb at 520 nm whereas it has been shown that the contribution of BrC to absorption at this wavelength can be high (e.g. Cuesta-Mosquera et al. (2023)). As a consequence, other studies (e.g. Zhang et al. (2019); Luo et al. (2022)) used the AAE calculated from 880 to 950 nm to calculate the $AAE_{BC}$ assuming that BrC particles do not absorb in the near IR. Nevertheless, the latter methodology may suffer from additional uncertainties related to the possible low aethalometer signal at 950 nm, frequently observed especially at remote sites. Thus, it should be considered that the methodologies proposed to estimate $AAE_{BC}$, including the use of the 1st percentile applied here, are prone to uncertainties. On the other hand, Zhang et al. (2020) have reported an uncertainty of approximately 11% in the estimation of the $b_{Abs,BrC(370)}$ contribution to $b_{Abs,370}$ when using different AAE values ranging from 0.9 and 1.1. For the sites included here, the 1st percentile method provides $AAE_{BC}$ values ranging from 0.928 to 1.088 confirming that this experimental method can provide reasonable estimations of the $AAE_{BC}$."

**References**

Blanco-Alegre, C., Fialho, P., Calvo, A., Castro, A., Coz, E., Oduber, F., Prevot, A., Močnik, G., Alves, C., Giardi, F., et al.: Contribution of coal combustion to black carbon: Coupling tracers with the aethalometer model, Atmospheric Research, 267, 105 980, 2022.

Bond, T. C., Doherty, S. J., Fahey, D. W., Forster, P. M., Berntsen, T., DeAngelo, B. J., Flanner, M. G., Ghan, S., Kärcher, B., Koch, D., et al.: Bounding the role of black carbon in the climate system: A scientific assessment, Journal of geophysical research: Atmospheres, 118, 5380–5552, 2013.

Cuesta-Mosquera, A., Glojek, K., Močnik, G., Drinovec, L., Gregorič, A., Rigler, M., Ogrin, M., Romshoo, B., Weinhold, K., Merkel, M., et al.: Optical properties and simple forcing efficiency of the organic aerosols and black carbon emitted by residential wood burning in rural Central Europe, EGUsphere, 2023, 1–34, 2023.

Glojek, K., Thuy, V. D. N., Weber, S., Uzu, G., Manousakas, M., Elazzouzi, R., Džepina, K., Darfeuil, S., Ginot, P., Jaffrezo, J., et al.: Annual variation of source contributions to PM10 and oxidative potential in a mountainous area with traffic, biomass burning, cement-plant and biogenic influences, Environment international, p. 108787, 2024.

Gyawali, M., Arnott, W., Lewis, K., and Moosmüller, H.: In situ aerosol optics in Reno, NV, USA during and after the summer 2008 California wildfires and the influence of absorbing and non-absorbing organic coatings on spectral light absorption, Atmospheric Chemistry and Physics, 9, 8007–8015, 2009.

Lack, D. and Langridge, J.: On the attribution of black and brown carbon light absorption using the Ångström exponent, Atmospheric Chemistry and Physics, 13, 10 535–10 543, 2013.

Li, L., Dubovik, O., Derimian, Y., Schuster, G. L., Lapyonok, T., Litvinov, P., Ducos, F., Fuertes, D., Chen, C., Li, Z., et al.: Retrieval of aerosol components directly from satellite and ground-based measurements, Atmospheric Chemistry and Physics, 19, 13 409–13 443, 2019.

Liakakou, E., Stavroulas, I., Kaskaoutis, D. G., Grivas, G., Paraskevopoulou, D., Dumka, U. C., Tsagkaraki, M., Bougiatioti, A., Oikonomou, K., Sciare, J., et al.: Long-term variability, source apportionment and spectral properties of black carbon at an urban background site in Athens, Greece, Atmospheric environment, 222, 117 137, 2020.

Lu, Z., Streets, D. G., Winijkul, E., Yan, F., Chen, Y., Bond, T. C., Feng, Y., Dubey, M. K., Liu, S., Pinto, J. P., et al.: Light absorption properties and radiative effects of primary organic aerosol emissions, Environmental science & technology, 49, 4868–4877, 2015.

Luo, B., Kuang, Y., Huang, S., Song, Q., Hu, W., Li, W., Peng, Y., Chen, D., Yue, D., Yuan, B., et al.: Parameterizations of size distribution and refractive index of biomass burning organic aerosol with black carbon content, Atmospheric Chemistry and Physics, 22, 12 401–12 415, 2022.

Qin, Y. M., Tan, H. B., Li, Y. J., Li, Z. J., Schurman, M. I., Liu, L., Wu, C., and Chan, C. K.: Chemical characteristics of brown carbon in atmospheric particles at a suburban site near Guangzhou, China, Atmospheric Chemistry and Physics, 18, 16 409–16 418, 2018.

Tian, J., Wang, Q., Ma, Y., Wang, J., Han, Y., and Cao, J.: Impacts of biomass burning and photochemical processing on the light absorption of brown carbon in the southeastern Tibetan Plateau, Atmospheric Chemistry and Physics, 23, 1879–1892, 2023.

Tobler, A. K., Skiba, A., Canonaco, F., Močnik, G., Rai, P., Chen, G., Bartyzel, J., Zimnoch, M., Styszko, K., Nęcki, J., et al.: Characterization of non-refractory (NR) PM 1 and source apportionment of organic aerosol in Kraków, Poland, Atmospheric chemistry and physics, 21, 14 893–14 906, 2021.

Via, M., Minguillón, M. C., Reche, C., Querol, X., and Alastuey, A.: Increase in secondary organic aerosol in an urban environment, Atmospheric Chemistry and Physics, 21, 8323–8339, 2021.

Wang, J., Nie, W., Cheng, Y., Shen, Y., Chi, X., Wang, J., Huang, X., Xie, Y., Sun, P., Xu, Z., et al.: Light absorption of brown carbon in eastern China based on 3-year multi-wavelength aerosol optical property observations and an improved absorption Ångström exponent segregation method, Atmospheric Chemistry and Physics, 18, 9061–9074, 2018.

Zhang, G., Peng, L., Lian, X., Lin, Q., Bi, X., Chen, D., Li, M., Li, L., Wang, X., Sheng, G., et al.: An improved absorption Ångström exponent (AAE)-based method for evaluating the contribution of light absorption from brown carbon with a high-time resolution, Aerosol and Air Quality Research, 19, 15–24, 2019.

Zhang, Y., Albinet, A., Petit, J.-E., Jacob, V., Chevrier, F., Gille, G., Pontet, S., Chrétien, E., Dominik-Sègue, M., Levigoureux, G., et al.: Substantial brown carbon emissions from wintertime residential wood burning over France, Science of the Total Environment, 743, 140 752, 2020.

Zotter, P., Herich, H., Gysel, M., El-Haddad, I., Zhang, Y., Močnik, G., Hüglin, C., Baltensperger, U., Szidat, S., and Prévôt, A. S.: Evaluation of the absorption Ångström exponents for traffic and wood burning in the Aethalometer-based source apportionment using radiocarbon measurements of ambient aerosol, Atmospheric Chemistry and Physics, 17, 4229–4249, 2017.

---

## Author Response (AR1)

**Response to anonymous referee #1: *Characterization of Brown Carbon absorption in different European environments through source contribution analysis**

November 18, 2024

**Major comments**

**Comment 1:** This is a very long paper. I would recommend that you go through to confirm that everything is needed in the main text or in the paper in general to make your points. You have an SI, but it's quite short, especially relative to the long main text.

**Response 1:** We appreciate your feedback regarding the length of the manuscript and the balance between the main text and the Supplementary Information (SI). We acknowledge that some sections could be streamlined without compromising the scientific message. The revised manuscript has been significantly shortened and specific material moved to the SI, now named Supplement which has been significantly extended.

Specifically, Section 3.1 has been shortened by moving the detailed discussion of the statistical evaluation to the Supplement and leaving the main message in the manuscript. Similarly, Section 3.2 has also been significantly shorten by moving the discussion of the optimization analysis of Cases 1 to 4 to the Supplement and keeping in the manuscript the analysis of Case 4 and Case 5, which are the relevant cases of the study. Finally, we have merged Sections 3.3.1 and 3.3.2 in a single and shorter section highlighting the main findings and implications of the derived refractive indexes. Some figures are now in the Supplement and we have introduced a simple and more descriptive Figure 5 that easily synthesis the impact of the constrained refractive index in calculating absorption in the model.

We believe that the revise manuscript is more easy to read and the main findings better highlighted.

**Comment 2:** Figure 3 could probably be better communicated in words or the figure could go in the SI.

**Response 2:** We acknowledge the suggestion of the reviewer. The purpose of Figure 3 was to schematize the flow diagram of the conversion process from organic aerosol (OA) mass to OA absorption, which involves multiple steps. We believe that presenting this process visually helps clarify the methodology and enhances understanding for the readers. However, we agree with the reviewer that it is not a fundamental figure in the main mansucript and it has been moved to the Supplement. The figure is cited in the main text as *Figure S1*.

**Comment 3:** Similarly Table 3 should be in the SI and then referenced.

**Response 3:** We thank the reviewer for their suggestion. After thorough consideration, we prefer to maintain Table 3 in the main manuscript as it provides the definition of the cases discussed throughout the manuscript. The table is also used in several sections of the document as a reference for the definition of the different absorption categories. We consider that this information is easily communicated through the Table and represents the core of part of the work.

**Comment 4:** Related to Table and Figure 4, can you help the reader better understand any trends you are attempting to make across sites? You are showing a lot of data, so I wonder if it might be more compelling to focus your main figures on large takeaways and then put these thumbnail plots in the SI. What are the big overarching takeaways that you want the reader to know about modeling emissions/absorption across locations? Across seasons? Etc.?

**Response 4:** We appreciate the feedback of the reviewer. Following your suggestion, we have moved Table 4 to the Supplement and revised the main text to emphasize the broader trends and key takeaways across sites and seasons, rather than overwhelming the reader with detailed site-specific plots. Now, a detailed analysis can be found in Section S3 in the Supplement leaving the main takeaways in the main manuscript.

In particular, we focus on the most significant findings related to OA mass concentrations and source contributions, specifically highlighting the following:

- Seasonal trends: The dominance of residential emissions during winter months across all stations, driven by heating demand, and the importance of Secondary Organic Aerosols (SOA) during the warmer months, which is particularly evident in summer with peaks at sites like HYY and MSY.

- Spatial variability: Different contributions from emission sources across locations, such as the significant role of shipping emissions at coastal stations like BCN and MAR, versus agricultural emissions at inland stations like IPR and MSY.

- Model performance: The model demonstrates strong capability in capturing seasonal patterns, showing good alignment between modeled and observed concentrations. But faces difficulties with specific cases such as underestimating SOA contributions in summer or overestimating residential emissions in certain Mediterranean regions during winter.

Thank you for encouraging us to clarify and simplify our presentation. We believe this revision makes the key findings more accessible while still providing comprehensive data.

**Comment 5:** Perhaps instead of so many figures with separate thumbnails for each site, think through some interesting summary figures that help people pull out the main takeaways.

**Response 5:** Thank you for your valuable suggestion. In response, we have revised the figures to provide more insightful summary visuals that emphasize the main takeaways. Specifically, we have made the following changes:

- We moved the original scatter plot figures, Figure 6 (now Figure S4), Figure 9 (now Figure S5), and Figure 10 (now Figure S6), to the supplementary material.

- Additionally, we created a new Figure 5, which consolidates the data from Cases 4 and 5 across all stations and throughout the year. This figure now contains four scatter plots showing the monthly mean results that summarize the key trends, providing a clearer representation of the results across different stations, environments, and methodological approaches used.

We believe that these changes will make it easier for readers to extract the insights from the study, as the new summary figure reduces redundancy and offers a comprehensive view of the data.

Thank you again for your helpful feedback. We believe that this revision addresses your concerns.

**Comment 6:** If you are going to keep Figure 8, then it should be better saturated to show differences. It's all blues and greens and not touching the top of your color bar.

**Response 6:** Thank you for your feedback on Figure 8. In response to your suggestion, we have modified the color scale of the figure to emphasize the differences. The updated figure is now Figure in the revised manuscript.

**Comment 7:** If Figure 9 is meant to be compared to Figure 6, then perhaps a summary figure that helps the reader see the comparison would be more useful. I am not fully convinced that the current figure is needed. Is your main point that the derived k performs better than an average one? Make that point in one sentence and then cite to a statistic showing that, and if you really desire, put this plot in the SI.

**Response 7:** Thank you for your comment. We agree that the comparison between Figure 6 and Figure 9 could be more effectively summarized. In response, we have moved the original scatter plots, Figure 6 (now Figure S4) and Figure 9 (now Figure S5), to the supplementary material.

To address your suggestion for a more concise and comparative visualization, we have replaced these figures with a new Figure 5, which provides a panel plot. This figure presents the monthly mean for Case 4 and Case 5 across all stations, differentiating between the two approaches (derived $k$ at each station vs. derive $k$ combining all data). This new summary plot clearly highlights the performance differences between these two approaches. This is now explicitly stated in the results section, supported by relevant statistics.

We believe that this approach better communicates the comparison while streamlining the presentation.

**Comment 8:** For Figure 10, similar points to above about Figure 9.

**Response 8:** Thank you for your comment regarding Figure 10. In line with your suggestions for Figure 9, we have moved Figure 10 to the supplementary material, where it is now labeled as Figure S6. This decision was made to streamline the main text and focus on more essential summary figures.

We believe this change enhances the clarity and flow of the manuscript while keeping the detailed data available for readers in the supplementary section.

**Comment 9:** In the summary and conclusions, you don't need to go over your major methods in a ton of detail again. Make your main points clearly and efficiently.

**Response 9:** We thank the reviewer for their suggestion to shorten the main text and presenting the main findings clearly and efficiently. Following your suggestion, we have shorten the summary and conclusions section removing the details from the first three paragraphs presented now in a single initial paragraph and highlighting the main findings and relevant conclusions in a synthesized and clearer way in this section.

**Minor comments**

**Comment 1:** Wording could also be more concise. For example line 27 ("BrC is originating") should be BrC originates.

**Response 1:** Thank you for pointing out this language improvement. We have now corrected this in the manuscript and used a more concise style.

**Comment 2:** For Figure 1, I would define BrC and BC with the colors that you're using in the plot instead of labeling all of them in black to the side of each pie chart.

**Response 2:** In Figure 1, we now define BrC and BC using the colors that correspond to each component in the plot, brown for BrC and black for BC, and remove the labeling in each pie chart. This change improves the clarity and visual consistency of the figure.

**Comment 3:** In Table 5 and Figure 5, please give the cases descriptive names so that the reader can better follow along.

**Response 3:** Thank you for your suggestion to provide descriptive names for each case in Table 5 (no Table S4) and Figure 5 (now Figure 4 in the revised manuscript). We understand that more descriptive titles could help readers follow more easily. However, each of the cases in our study represents a unique combination of multiple categories from different sources, making it challenging to assign concise yet informative names to each case.

After considering your feedback, we attempted various naming options, but found that any descriptive names we tried to assign either became overly complex or failed to capture the unique attributes of each case. Consequently, we decided to retain the labels "Case 1" through "Case 5" for simplicity and clarity and refer the reader to Table 3 for their exact definition.

**References**

**Response to anonymous referee #2: *Characterization of Brown Carbon absorption in different European environments through source contribution analysis**

November 18, 2024

**Major comments**

**Comment 1:** The manuscript is too long. Please consider avoiding such long papers in the future.

**Response 1:** We appreciate your feedback regarding the length of the manuscript. We acknowledge that some sections could be streamlined without compromising the scientific message. The revised manuscript has been significantly shortened and specific material, including figures and tables, moved to the Supplement. We believe this revision makes the key findings more accessible while still providing comprehensive information.

Specifically, Section 3.1 has been shortened by moving the detailed discussion of the statistical evaluation to the Supplement and leaving the main message in the manuscript. Similarly, Section 3.2 has also been significantly shorten by moving the discussion of the optimization analysis of Cases 1 to 4 to the Supplement and keeping in the manuscript the analysis of Case 4 and Case 5, which are the relevant cases of the study. Finally, we have merged Sections 3.3.1 and 3.3.2 in a single and shorter section highlighting the main findings and implications of the derived refractive indexes. Some figures are now in the Supplement and we have introduced a simple and more descriptive Figure 5 that easily synthesis the impact of the constrained refractive index in calculating absorption in the model.

**Comment 2:** For OA mass concentration, how do you account for the collection efficiency?

**Response 2:** The OA mass concentrations measured with Aerosol Chemical Speciation Monitor (ACSM) instruments reported here were published in Chen et al. (2022) in the framework of the COLOSSAL Cost Action (CA16109 Chemical On-Line cOmpoSition and Source Apportionment of fine aerosoLs) (Freney et al., 2019). As reported in Freney et al. (2019), a composition dependent collection efficiency was calculated for each instrument and applied to each dataset following the recommendations by Middlebrook et al. (2012).

To clarify the reviewer's comment, the sentence at Lines 121-124 was modified as follow:

"Details about ACSM instruments used for OA determination, measurement principle, accuracy and treatment of sources of error as collection efficiency can be found in Ng et al. (2011), Middlebrook et al. (2012), Fröhlich et al. (2013), Freney et al. (2019) and Chen et al. (2022)."

**Comment 3:** For equations 1 and 2, it should be noted that many current studies show BrC can also absorb at NIR ("Optical Properties of Individual Tar Balls in the Free Troposphere", "Shortwave absorption by wildfire smoke dominated by dark brown carbon", and "Brown carbon absorption in the red and near-infrared spectral region"). Thus, assuming only BC absorbs light

at 880 nm will underestimate the babs of BrC. I suggest adding some relevant discussions. This might explain why your k is so low.

**Response 3:** In order to consider the Reviewer comment, the following sentence below equations 1 and 2 (from line 152 to line 171 in the revised manuscript) was accordingly modified as follows:

" where $AAE_{BC}$ is the Absorption Angstrom Exponent (AAE) of BC, which allows calculating $b_{abs,BC(\lambda)}$ (in units of $Mm^{-1}$) from the measurements of $b_{abs,BC}$ at 880 nm assuming that BrC does not absorb at 880 nm (e.g., Qin et al. (2018)). It should be noted, however, that recent studies have shown the existence of specific dark BrC components in biomass-burning (BB) smoke (tar balls or tar BrC) that can absorb radiation also in the near-infrared (e.g. Chakrabarty et al. (2010); Hoffer et al. (2016, 2017); Chakrabarty et al. (2023); Mathai et al. (2023)). Thus, a contribution to near-IR absorption from possible presence of dark BrC cannot be ruled and would lead to an underestimation of the BrC absorption reported here. However, the dark BrC contribution to absorption at 880 nm is expected to be smaller compared to that of BC. For example, Hoffer et al. (2017) reported that the absorption coefficient at 880 nm of dark BrC produced in a laboratory was 10% of that at 470 nm and, consequently, even lower compared to that at 370 nm. Similarly, Cuesta-Mosquera et al. (2023) estimated a contribution of BrC to absorption at 880 nm of 3% in a rural area in central Europe strongly affected by residential wood burning emissions in winter. Given the complexity of these specific BrC components, the imaginary refractive index (k) of tar balls generated in laboratory experiments vary over a wide range of values depending on the specific type of fuel (wood) burned and the different analytical methods employed. For example, Mathai et al. (2023) reported k values at 550 nm of tar balls measured in ambient BB plumes 10 times lower than the values reported by Hoffer et al. (2016), Chakrabarty et al. (2010) or Saleh et al. (2018); Saleh (2020) for laboratory generated particles. Also, Mathai et al. (2023) highlighted that even though Hoffer et al. (2016) and Chakrabarty et al. (2010) used similar methods, their k was at least 10 times different. Thus, due to poorly characterized optical properties, the impact of tar BrC on IR absorption at ambient conditions is still uncertain. Consequently, we follow here the common practice of considering ambient BC as the dominant absorber at 880 nm (e.g. Kirchstetter et al. (2004); Massabò et al. (2015); Liakakou et al. (2020); Zhang et al. (2020); Yus-Díez et al. (2022)). One important source of uncertainty in equations 1 and 2 is the AAE assumed for BC."

**Comment 4:** It is not clear to me why you chose 370 nm instead of 550 nm for the discussion of light absorption, which is widely used for discussing aerosol optical properties. Could you justify that?

**Response 4:** We agree with the reviewer that the wavelength of 550 nm is commonly used to discuss the optical properties of atmospheric aerosol particles, especially for the scattering part given the high efficiency of submicrometric PM to scatter visible radiation. The same 550 nm wavelength has been also used to represent the absorption (e.g. Samset et al. (2018); Saleh et al. (2018); Saleh (2020)). Our decision to report here the OA absorption at 370 nm was due to the main fact that the OA absorption efficiency is the highest at 370 nm and for this reason the 370 nm wavelength is widely used in many publications reporting OA absorption from Aethalometer measurements. Consequently, an advantage of representing the OA absorption at 370 nm is that it can be retrieved from observations with a lower uncertainty compared to the visible (550 nm) range.

Moreover, for the k optimization we started from the $k$ values proposed by Saleh (2020) and we used the Angstrom exponents (AAE) proposed by Saleh (2020) to report the $k$ at 370 nm. Again, this choice was made in order to compare the modelled OA absorption with the observed

OA absorption at 370 nm with lower uncertainty. The same AAE provided by Saleh (2020) could be used to derive the optimized $k$ reported here from 370 nm to 550 nm.

To consider the Reviewer comment, the sentence at lines 173-175 was modified as follow (now lines 200-204 in the revised manuscript):

"Figure 1 shows the average annual contributions of BC and BrC to the total absorption measured at 370 nm at twelve European stations, identified by color-coded markers indicating their background settings: yellow for urban, blue for suburban, and red for regional areas. Here we report the BrC absorption at 370 nm given that the BrC absorption efficiency is the highest in the UV spectral range and, consequently, the observed BrC absorption is less uncertain compared to the visible range."

**Comment 5:** Your k at and babs at 370 nm seems too low to me. some babs even lower than some literature values reported from Arctic (see "On Aethalometer measurement uncertainties and an instrument correction factor for the Arctic" and "On Aethalometer measurement uncertainties and an instrument correction factor for the Arctic").

**Response 5:** The absorption reported in Figure 6 of the revised manuscript represents the contribution by OA particles to the total absorption at 370 nm. Thus, it represents a fraction of the total absorption measured at each site. Indeed, in summer, the reported OA absorption reached values lower than 1 $Mm^{-1}$ especially at more remote sites. However, the OA contribution to absorption in summer is expected to be low due to the lack of important primary BrC sources as domestic biomass burning.

Moreover, here we used the values of $k$ provided by Saleh (2020) at 550 nm. These $k$ at 550 nm were calculated at 370 nm using the w Angstrom exponents provided by Saleh (2020). If, after the $k$ optimization at 370 nm (Table S3 in the revised Supplement), the original w Angstrom exponents from Saleh (2020) were used to report the optimized $k$ from 550 nm at the 370 nm, these latter will lie somewhere within the same original ranges provided by Saleh (2020).

To consider this Reviewer comment, the sentence at lines 590-591 (lines 535-540 in the revised manusrcipt) was modified as follow:

"Additionally, Figure 6 shows the time series of the absorption of OA at 370 nm simulated in Case 4 (all) and Case 5 (all) and the observational data for each monitoring site (by 'stn' in Fig. S6 in the Supplement). Although the absorption values could seem low, the annual mean OA absorption at 370 nm calculated from observations represented around 2-20% of the total measured absorption at some regional/remote sites (MSY, OPE and HYY) reaching contributions around 20-40% at the remaining sites. The OA absorption is the lowest in summer due to the lack of important primary BrC sources as domestic biomass burning. "

**Specific comments**

**Comment 1:** Line 175-183, The discussion of BC and BrC sources lacks the evidence. Do you make these conclusions only based on the contribution? Or can any references support you? Or other data was involved.

**Response 1:** To consider the Reviewer comment, the sentence at Lines 175-183 (lines 200-219 in the revised manuscript) was modified as follow:

Figure 1 shows the average annual contributions of BC and BrC to the total absorption measured at 370 nm at twelve European stations, identified by color-coded markers indicating their background settings: yellow for urban, blue for suburban and red for regional areas. Here we report the BrC absorption at 370 nm given that the BrC absorption efficiency is the highest in the UV spectral range and, consequently, the observed BrC absorption is less uncertain compared

to the visible range. A low BrC contribution, around 14%, was observed at the urban sites of BCN and HEL, both affected by direct traffic emissions, making BC the dominant absorber at these sites (Okuljar et al., 2023; Via et al., 2021). MAR urban site registered higher BrC contribution (30%) likely reflecting the accumulation of biomass burning emissions in winter and the presence of BrC sources as shipping emissions (Corbin et al., 2018; Chazeau et al., 2022). Suburban stations, including SIR, KRA, and DEM, exhibit BrC contributions from 22% to 30%, reflecting a blend of local urban emissions and regional influences such as biomass burning and coal combustion. KRA is considered a pollution hotspot in Europe (e.g. Casotto et al. (2023)) with high consumption of coal and wood (both important sources of BrC) for energy production and residential heating, making the OA concentrations measured in KRA the highest among the European measuring sites included in Chen et al. (2022). DEM and SIR are affected by biomass burning emissions especially in winter, which causes a considerable accumulation of BrC during the cold season (see, e.g., Liakakou et al. (2020); ?); Savadkoohi et al. (2023)). Regional stations, represented by HYY, OPE, RIG, PAY, IPR, and MSY, display BrC contributions from 21% to 41%. These percentages indicate a mixture of biogenic sources, local emissions, agricultural activities, and transboundary pollution that affects the regional atmosphere. IPR stands out with the highest contribution (around 40%), suggesting a significant contribution of low-temperature combustion processes as residential sources (e.g. Putaud et al. (2018)). Overall, although BC typically represents the most absorbing aerosols component at these stations (usually $> 70\%$), it is noteworthy that BrC could contribute comparably to absorption in some instances.

**Comment 2:** Line 214-215. It is not obvious to me how you make these assumptions. Why do you assume 50% hydrophobic species with OA/OC = 1.4 and all hydrophilic oxygenated components have OA/OC=2.1? How about other 50% hydrophobic species?

**Response 2:** There might be a misunderstanding in the description of the model's parameterization. Various approaches exist to represent organic aerosols (OA) in atmospheric models. In this study, we use the parameterization proposed by Pai et al. (2020). Typically, models treat primary organic aerosols as one hydrophobic and one hydrophilic component (Xian et al., 2019). This approach aims to simulate the near-field oxidation of the primary hydrophobic component. Emission inventories provide estimates of total organic carbon (OC) emissions, requiring assumptions to convert it to OA mass, the variable actually used in atmospheric transport models. In Pai et al. (2020)'s parameterization, OC emissions split equally between hydrophobic and hydrophilic aerosols. As MONARCH transports OA mass, it requires an assumption for OC-to-OA. Literature suggests ratios from 1.4 to 1.6 for hydrophobic OA species and over 2 for oxygenated species, depending on the combustion characteristics. Such values are highly uncertain and slightly different numbers are used among model parameterizations. As stated before, in this work we follow the parameterization proposed by Pai et al. (2020).

To clarify this point, the description of the scheme has been modified in the revised manuscript as follow:

Primary organic carbon (OC) emissions are emitted as 50% hydrophobic and 50% hydrophilic species. An OA/OC ratio of 1.4 is adopted for the hydrophobic component, while the hydrophilic one assumes an OA/OC ratio of 2.1 to convert OC to OA mass transported in the model.

**Comment 3:** Line 215-216: It is unclear how you get the conversion lifetime of 1.15 days.

**Response 3:** As explained in the previous comment, we adopted an OA parameterization proposed in the literature by Pai et al. (2020). The atmospheric aging of primary hydrophobic particles is simulated by its conversion to hydrophilic aerosol with a specific atmospheric lifetime.

This parameterization uses a lifetime of 1.15 days as previously proposed by other studies (Chin et al., 2002; Cooke et al., 1999).

**Comment 4:** Do you have references for these values in table 2, or did you derived them?

**Response 4:** The aerosol representation used in the MONARCH model has been described elsewhere (Spada, 2015; Obiso, 2018; Navarro-Barboza et al., 2024). The microphysical properties of the organic aerosol follow Chin et al. (2002), which in turn is based on the OPAC database (Hess et al., 1998). In order to clarify this point, Table 2 caption of the revised manuscript now includes references to Spada (2015); Chin et al. (2002); Hess et al. (1998).

**References**

[revised manuscript text omitted]